# TM9SF4 is an F-actin disassembly factor that promotes tumor progression and metastasis

Zhaoyue Meng[1,2,4], Zhichao Li[1,3,4], Mingxu Xie[1,4], Hongyan Yu[1], Liwen Jiang ®[2] ✉ & Xiaoqiang Yao ®[1,2] ✉

F-actin dynamics is crucial for many fundamental properties of cancer cells, from cell-substrate adhesion to migration, invasion and metastasis. However, the regulatory mechanisms of actin dynamics are still incompletely understood. In this study, we demonstrate the function of a protein named TM9SF4 in regulating actin dynamics and controlling cancer cell motility and metastasis. We show that an N-terminal fragment (NTF) cleaved from TM9SF4 can directly bind to F-actin to induce actin oxidation at Cys374, consequently enhancing cofilin-mediated F-actin disassembly. Knockdown of TM9SF4 reduces cell migration and invasion in ovarian cancer cells A2780, SKOV3 and several high grade serous ovarian cancer lines (HGSOCs). In vivo, knockdown of TM9SF4 completely abolishes the tumor growth and metastasis in athymic nude mice. These data provide mechanistic insights into TM9SF4-mediated regulation of actin dynamics in ovarian cancer cells.

The rapid assembly and disassembly of F-actin cytoskeleton underlies diverse cell behaviours, including growth, migration and phagocytosis[1]. In cancer cells, F-actin dynamics, namely its assembly and disassembly, is crucial for many fundamental cell properties, from cell-substrate adhesion to migration and invasion[2]. Actin assembly involves addition of G-actin at the barbed end of F-actin, while actin disassembly occurs at the pointed end of actin filament driven by ADF/cofilin[3,4]. ADF/cofilin contributes to actin dynamics by severing F-actin to promote actin disassembly. Severing of F-actin by ADF/cofilin also creates a free barbed end which is required for subsequent actin polymerization[4]. However, the regulatory mechanisms of actin dynamics are still incompletely understood.

F-actin dynamics can be regulated by post-translational modification (PTM)[5]. Among the different types of PTM, oxidation-reduction (Redox) modification of amino acid residues has emerged as a key modulator of actin cytoskeleton dynamics[6]. Two types of redox enzymes, NADPH oxidases (NOX) and Mical flavoprotein monooxygenases, are well-documented to regulate actin dynamics. The NOX family comprises 7 members, including NOX1–5 and Duox1–2[7]. NOX produces reactive oxygen species to induce oxidation of F-actin, consequently attenuating actin polymerization[6]. As for Mical, it can bind physically to F-actin. Such binding stimulates the enzymatic oxidase activity of Mical, consequently facilitating cofilin-mediated severing and disassembly of F-actin[8,9]. Functionally, the NOX- and Mical-mediated oxidation of F-actin have been shown to regulate cell migration and axon guidance[8,9].

The TM9SF family is characterized by a long hydrophilic N terminus and nine putative transmembrane domains in the C-terminal region[10]. The TM9SF family is highly conserved through evolution, being expressed in *Dictyostelium discoideum, Saccharomyces cerevisiae, Arabidopsis thaliana, Drosophila melanogaster* and mammals[11,12]. In mammals, there are four different members (TM9SF1–TM9SF4)[11]. Functional studies showed that TM9SF4 plays an important role in phagocytosis, autophagy and cancer cell cannibalistic activity[11,13–16]. Mutations in TM9SF4 result in alteration of cell morphology, including an increased cell size and aberrant actin spikes in *Drosophila* macrophages[13]. While it is tempting to suggest that the above-mentioned cell behaviors and morphological changes could be

[1]School of Biomedical Sciences and Li Ka Shing Institute of Health Sciences, Faculty of Medicine, The Chinese University of Hong Kong, Hong Kong, China. [2]Centre for Cell & Developmental Biology and State Key Laboratory of Agrobiotechnology, School of Life Sciences, The Chinese University of Hong Kong, Hong Kong, China. [3]Key Laboratory of Medical Reprogramming Technology, Shenzhen Second People's Hospital, The First Affiliated Hospital of Shenzhen University, Shenzhen, China. [4]These authors contributed equally: Zhaoyue Meng, Zhichao Li, Mingxu Xie. ✉e-mail: ljiang@cuhk.edu.hk; yao2068@cuhk.edu.hk

linked to F-actin dynamics, up to the present there is no direct evidence showing that TM9SF4 can regulate F-actin dynamics. Interestingly, the TM9SF family of proteins exhibits NADH oxidase activity with its catalytic site located in the N-terminal region[17]. Reports show that an N-terminal fragment of ~30 kDa can be shed from the full length TM9SF into body fluid (serum, urine and saliva) at old age[18]. However, it is still unclear whether TM9SF4 can regulate F-actin dynamics by its oxidase activity.

In the present study, our results demonstrate that knockdown of TM9SF4 increases the formation of actin stress fibers, while overexpression of an N-terminal fragment (NTF) of TM9SF4 disrupts the actin filaments. Mechanistic studies showed that TM9SF4 facilitates the cofilin-induced disassembly of F-actin and that knockdown of TM9SF4 markedly inhibits cancer cell migration and invasion. Importantly, TM9SF4 depletion completely abolishes the ovarian cancer metastasis in athymic nude mice. Together, this study provides evidence that TM9SF4 can facilitate the cofilin-mediated disassembly of F-actin by redox regulation, consequently regulating cancer cell migration in vitro and metastasis in mouse model in vivo.

## Results

### Knockdown of TM9SF4 promotes F-actin formation in tumor cells

The expressional level of TM9SF4 was compared among several cancer cell lines and noncancerous cell lines. Immunoblot results showed that the endogenous protein expression level of TM9SF4 was much higher in ovarian cancer lines, including A2780, CaOV3, OVCAR3, COV362 and SKOV3, and breast cancer lines MCF-7 and T47D cells, than in noncancerous lines NIH3T3 and MCF-10A cells (Fig. 1a and Fig. S1a).

We examined the effect of TM9SF4 knockdown on cell morphology and growth. Lentivirus-based TM9SF4-specific shRNAs (shTM9SF4 #1 and shTM9SF4 #2 for human cells, shTM9SF4 #3 and shTM9SF4 #4 for mouse cells) were constructed, each of which could effectively knockdown the expression of TM9SF4 in respective cell lines (Fig. 1b and Fig. S1b). TM9SF4 knockdown substantially increased cell size and enhanced actin stress fiber formation in A2780, CaOV3, OVCAR3, COV362, SKOV3, MCF-7 and T47D cells, all of which have high endogenous expression levels of TM9SF4 (Fig. 1c−e and Fig. S2), while the expression level of total actin remained unaffected (Fig. S3). However, in those cells which have a low endogenous expression level of TM9SF4, including MCF-10A and NIH3T3 cells, TM9SF4 knockdown had no obvious effect on cell morphology (Fig. 1c−e). In addition, TM9SF4 knockdown decreased the cell population growth in A2780, CaOV3, OVCAR3, COV362, SKOV3, MCF-7 and T47D cells, but not in MCF-10A and NIH3T3 cells (Fig. S4). Furthermore, normal A2780 cells only had one nucleus, while TM9SF4 knockdown resulted in unsuccessful cell division and the formation of multinuclear cells, suggesting a defect in cytokinesis due to abnormal actin assembly and disassembly (Fig. S5). The decreased cell population growth was mainly due to reduced cell proliferation as indicated by EdU staining (Fig. S6) with little contribution from cell death as measured by flow cytometer-based PI analysis (Fig. S7).

Further examination on cell morphology showed that TM9SF4 knockdown in A2780 and MCF-7 cells led to the formation of multipolar actin-rich protrusions (Fig. 1f, g and Fig. S8), suggesting a possible role of TM9SF4 in actin dynamics. We thus tested whether TM9SF4 knockdown would affect the balance between F-actin and G-actin. After ultracentrifugation of the cell lysates prepared from A2780, CaOV3 and COV362 cells with or without TM9SF4 knockdown, supernatant and pellet fractions were collected. The actin in the pellet represented F-actin while the actin in the supernatant represented G-actin. Compared with the negative control transfected with scrambled shRNA, knockdown of TM9SF4 reduced the G-actin/F-actin ratio (Fig. 1h, i and Fig. S9), suggesting that TM9SF4 knockdown favors F-actin assembly. As a positive control, cells were treated with

Jasplakinolide (Jpk), a toxin that promotes actin polymerization. As expected, Jpk treatment also reduced the G-actin/F-actin ratio (Fig. 1h, i and Fig. S9). Together, these data suggest that TM9SF4 knockdown promotes F-actin assembly.

### Identification of an N-terminal fragment of TM9SF4 in A2780 cells

TM9SF4 is a membrane protein with nine transmembrane domains and a long hydrophilic N-terminus with a signal peptide[10]. Previous studies reported that TM9SF can be cleaved to form a ~30 kDa N-terminal fragment (NTF)[18]. To confirm this, several TM9SF4 constructs containing different tags were used (Fig. 2a). We first overexpressed a fusion protein sp-GFP-TM9SF4, in which a sp-GFP was used to replace the signal peptide (sp) of TM9SF4, in A2780 cells. Under mild denaturing conditions (without boiling of the protein samples), immunoblotting with GFP antibodies detected a band at around 85 kDa (Fig. 2b), likely representing full length GFP-TM9SF4, because the reported molecular weights of TM9SF4 and GFP were ~60 kDa and 27 kDa, respectively[15]. Interestingly, a band at ~55 kDa was also observed (Fig. 2b), which likely represented GFP-TM9SF4 NTF, for the expected molecular weight of TM9SF4 NTF is ~30 kDa. Under the strong denaturing condition of protein sample boiling, the full length GFP-TM9SF4 aggregated, causing a shift of its corresponding band to the top of resolving gel, while the band representing GFP-TM9SF4 NTF remained unchanged (Fig. 2b). Similar results were also observed in TM9SF4-V5-expressing cells (Fig. S10b).

TM9SF4 was reported to be localized in the Golgi, ER, and lysosomes[15,19,20]. Here we confirmed its predominant location at Golgi, based on its colocalization with a Golgi marker GM130 in A2780 cells (Fig. 2c). To further investigate the cleavage of TM9SF4 in cells, we next overexpressed another fusion protein mCherry-TM9SF4-GFP, in which TM9SF4 was tagged with mCherry at its N-terminus and tagged with GFP at its C-terminus. The results showed that the full-length fusion protein (in yellow) was mainly localized in Golgi (Fig. 2d). Intriguingly, distinct puncta representing truncated mCherry-TM9SF4 NTF (in red) were found to be located separately from the full length mCherry-TM9SF4-GFP (Fig. 2d). Immunoblots with mCherry antibodies showed two distinct bands, representing mCherry-TM9SF4-GFP and mCherry-TM9SF4 NTF, respectively (Fig. S10a). Cell fractionation assay further demonstrated that the NTF was in cytosol fraction while the full-length TM9SF4 was in membrane fraction (Fig. 2e). Together, these data further support the notion that the full-length TM9SF4 can be cleaved to form a TM9SF4 NTF, which resides in the cytoplasm of A2780 cells.

### TM9SF4 NTF interacts with actin

To identify the interacting partners of TM9SF4, TM9SF4 proteins were immunoprecipitated followed by mass spec identification of its binding partners. Two cell lines, A2780 and NIH3T3, with or without GFP-TM9SF4 overexpression, were used in these experiments. Whole cell lysates of control cells or GFP-TM9SF4-expressing cells were immunoprecipitated with anti-TM9SF4 or GFP antibodies, respectively. Mass spectrometry analysis of the precipitated proteins revealed actin as one of the major TM9SF4-interacting proteins (Fig. S11 and Supplementary Data 1). To confirm this finding, co-immunoprecipitation was performed in GFP-TM9SF4-expressing A2780 cells. As expected, GFP antibodies could pull down actin. However, actin antibodies could only pull down a product of ~55 kDa (Fig. 2f), likely representing GFP-TM9SF4 NTF, suggesting that actin preferentially interacted with TM9SF4 NTF.

We next made an artificial TM9SF4 NTF (TM9SF4_N$^{1-258}$-GFP), which only contains N-terminal 258aa, to mimic the native TM9SF4 NTF. TM9SF4_N$^{1-258}$-GFP was transfected to NIH3T3 cells, which have low endogenous level of TM9SF4. Immunoblot analysis confirmed that overexpressed TM9SF4_N$^{1-258}$ has a molecular weight similar to that of

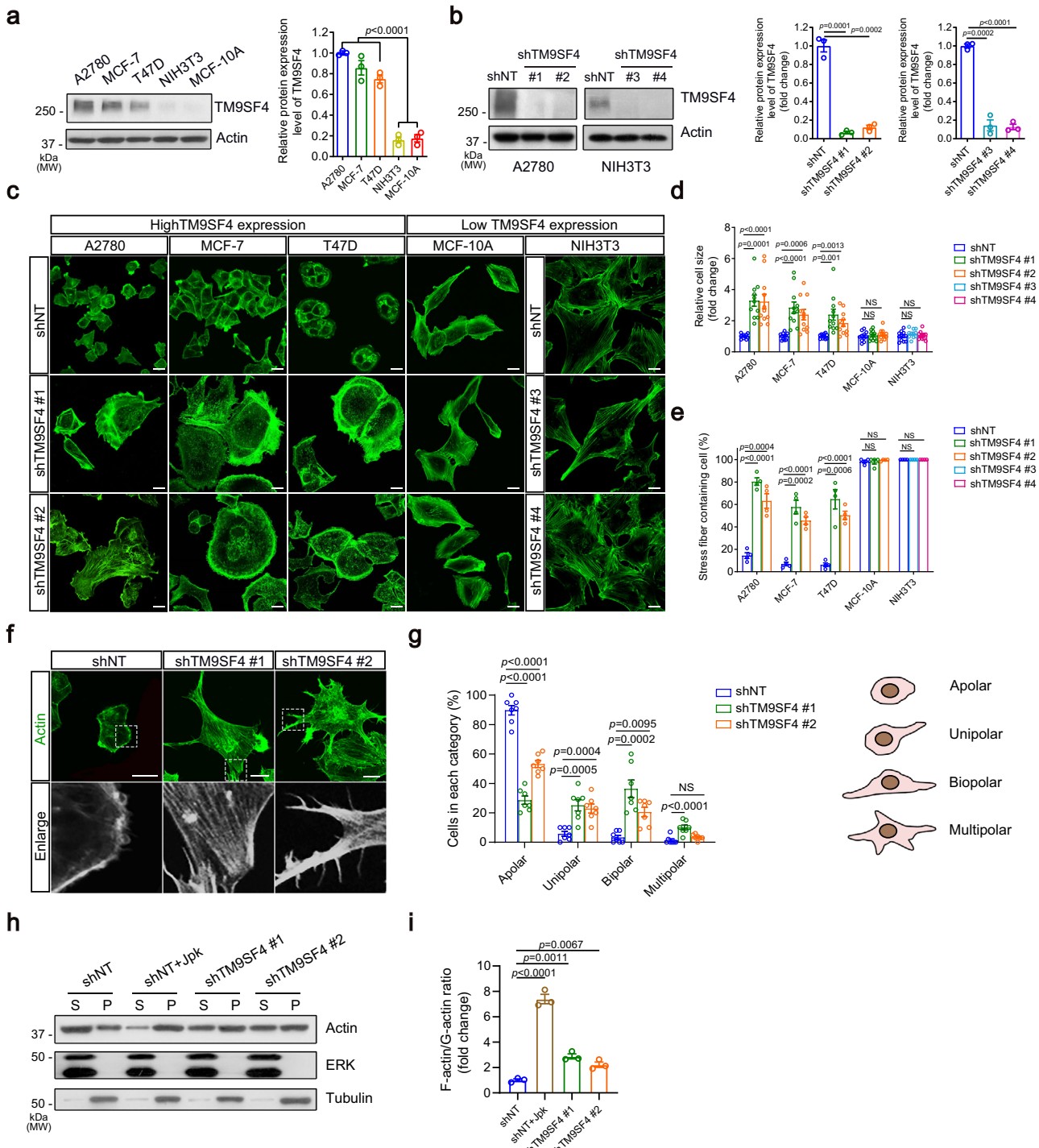

**Fig. 1 | Knockdown of TM9SF4 increases the cell size and promotes F-actin formation in cancer cells. a, b** Endogenous expression level of TM9SF4 in different cell types (**a**) and knockdown efficiency of TM9SF4-shRNAs (**b**). Shown are representative immunoblot images (left) and summary data (right) of TM9SF4 protein level (*n* = 3 biologically independent experiments). **c**–**e** TM9SF4 knockdown by lenti-shRNAs increased the cell size and F-actin stress fiber formation in A2780 cells, MCF-7 cells and T47D cells, but not in NIH3T3 cells and MCF-10A cells. Shown are representative confocal microscope images after FITC-phalloidin staining (**c**) and summary data (**d, e**). In **e**, the percentage of cells containing thick actin stress fiber with a diameter >0.5 μm was analyzed (*n* = 4 biologically independent experiments, >200 cells from three different fields per experiment). Scale bar = 10 μm. **f, g** TM9SF4 knockdown increased the formation of actin spikes and multipolar morphology in A2780 cells. Shown are representative cell images after FITC-phalloidin staining (**f**) and summary data (**g**) (*n* = 7 biologically independent experiments). Scale bar = 10 μm. **h, i** TM9SF4 knockdown increased the F-actin as in pellet (P) while reduced the G-actin as in supernatant (S) in A2780 cells. Shown are representative immunoblot images (**h**) and summary data (**i**). ERK, cytosolic marker; β-tubulin, cytoskeleton marker (*n* = 3 biologically independent experiments). Data are presented as mean ± SEM and two-tailed unpaired Student's *t*-test was used for statistical analysis (**a, b, d, e, g, i**). NS, no significant difference. Source data are provided as a Source Data file.

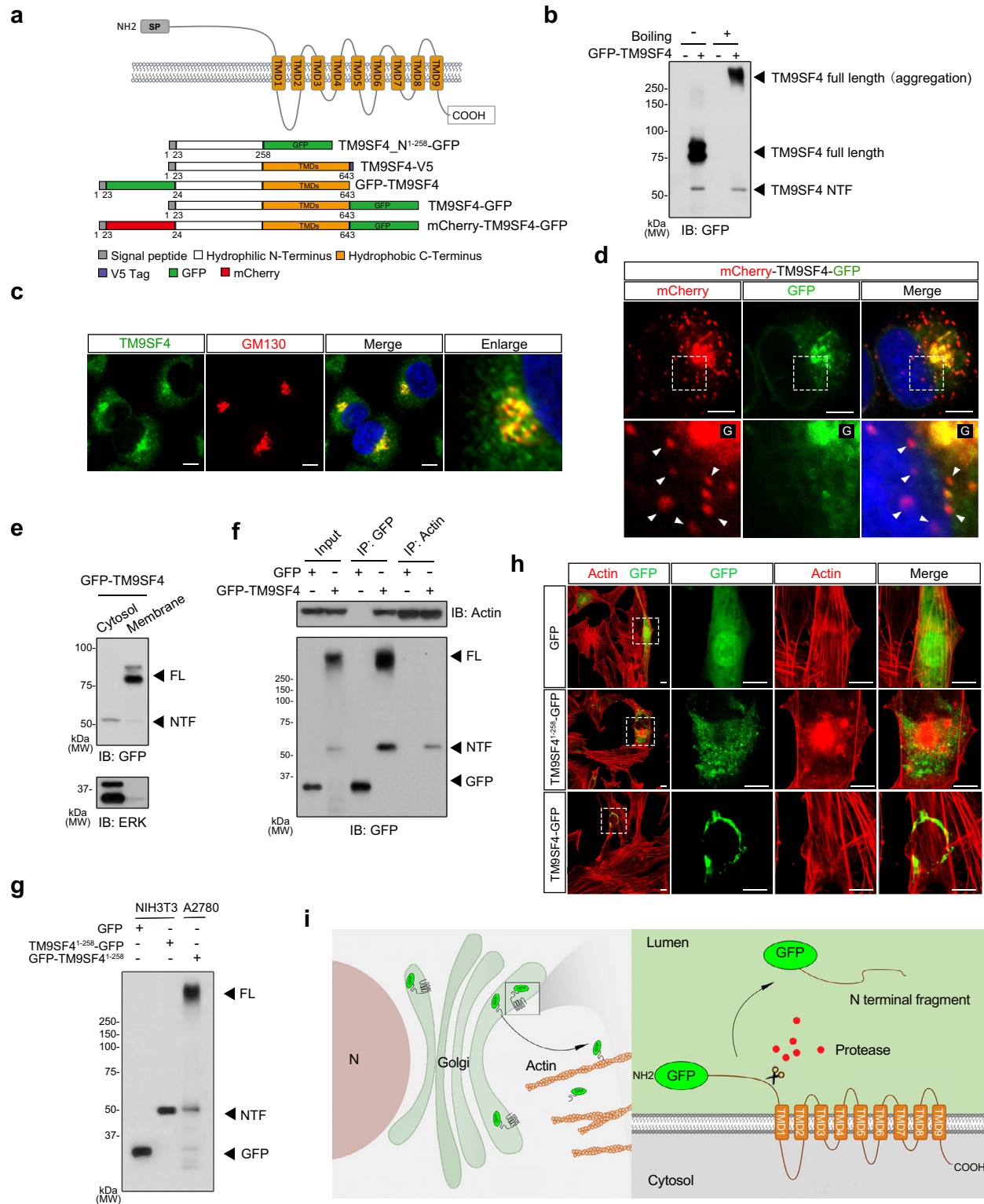

native TM9SF4 NTF, which was cleaved from the full length TM9SF4 (Fig. 2g). Intriguingly, overexpression of TM9SF4_N$^{1-258}$ caused a drastic disruption of F-actin fibers in NIH3T3 cells whereas overexpression of the full-length TM9SF4 did not show obvious alteration in the actin cytoskeleton (Fig. 2h). The possible reason is that full-length TM9SF4 could not be properly cleaved to form an N-terminal fragment in NIH3T3 cells (Fig. S12). A schematic figure is shown to illustrate the cleavage of TM9SF4 to form NTF in A2780 cells, which then interacts with actin (Fig. 2i).

## TM9SF4 NTF contains an actin binding domain and a redox domain

Bioinformatic analysis revealed that TM9SF4_N$^{1-258}$ contains a conserved actin binding domain (ABD, LKXXEX), that can be found in many actin-binding proteins such as PKC-ε and α-actinin[21,22], as well as a copper redox center (Fig. 3a). Different molecular constructs were generated to investigate the function of each domain (Fig. 3a).

To determine the actin binding site in TM9SF4 NTF, various GFP-tagged constructs were transfected in NIH3T3 cells, followed by

**Fig. 2 | The N-terminal fragment (NTF) of TM9SF4 interacts with actin.**
**a** Schematic illustration of full-length (FL) TM9SF4 and various TM9SF4 expression constructs with different tags. **b** Immunoblot analysis of whole cell lysates from GFP-TM9SF4 overexpressing A2780 cells. A GFP-TM9SF4 NTF was detected. **c** Subcellular colocalization of TM9SF4 proteins with Golgi marker GM130, as detected by respective antibodies. Scale bar = 5 μm. **d** Cleavage of overexpressed mcherry-TM9SF4-GFP to produce mcherry-TM9SF4 NTF in A2780 cells, as demonstrated by presence of mcherry distinct puncta (red). Arrow heads indicate the red puncta, located separately from the full length TM9SF4 (yellow). G, Golgi; Scale bar = 5 μm. **e** Subcellular fractionation followed by immunoblots showing the presence of TM9SF4 NTF in cytosol fraction and full length TM9SF4 in membrane fraction. GFP-TM9SF4 were overexpressed in A2780 cells. The blotting antibody was anti-GFP antibody. **f** Co-immunoprecipitation experiments showing that the anti-actin antibody could pull down TM9SF4 NTF in A2780 cells. The pulling antibody in immunoprecipitation (IP) and the blotting antibody in immunoblots (IB) were as labeled. **g** GFP and TM9SF4$^{41-258}$-GFP were transiently expressed in NIH3T3 cells while full-length GFP-TM9SF4 was transiently expressed in A2780 cells. TM9SF4$^{41-258}$-GFP construct expressed in NIH3T3 cells displayed the molecular weight similar to that of GFP-TM9SF4 NTF, a product truncated from GFP-TM9SF4 in A2780 cells. **h** GFP, TM9SF4$^{41-258}$-GFP and TM9SF4-GFP were transiently expressed in NIH3T3 cells. GFP signal was detected in each treatment and actin was visualized by rhodamine phalloidin staining (Red). Only TM9SF4$^{41-258}$-GFP disrupted the actin cytoskeleton. Scale bar = 5 μm. For (**b–h**), the experiments were repeated three times with similar results. **i** Schematic figure illustrating TM9SF4 cleavage in Golgi, followed by release into cytosol and interaction with actin cytoskeleton. Source data are provided as a Source Data file.

immunoprecipitation with GFP antibodies and immunoblots with actin antibodies (Fig. 3b). The results showed that, either TM9SF4_N$^{1-258}$-GFP or TM9SF4_N$^{178-258}$-GFP could co-immunoprecipitate with actin, whereas TM9SF4_N$^{1-177}$-GFP failed to do so (Fig. 3b), indicating that the aa178–258 is essential for actin binding. It was reported that the critical residues for actin binding are the conserved leucine and lysine residues (LK) which directly interact with actin[22]. We therefore converted the two key amino LK$^{220}$ to AA in the putative ABD of TM9SF4 NTF, the results showed that these mutations abolished the interaction between actin and TM9SF4 NTF (Fig. 3b). These data support LK$^{220}$XXEX in TM9SF4 as authentic actin binding site. Further evidence was provided by the subcellular localization of GFP-tagged TM9SF4 NTF and F-actin. The results showed that TM9SF4_N$^{178-258}$-GFP was well-colocalized with actin filaments, whereas mutation/depletion of ABD abolished the colocalization (Fig. S13). An in vitro actin co-sedimentation assay using purified proteins was performed to further confirm the actin binding site. The results showed that removal of amino acids 178–258 (GST-TM9SF4_N$^{24-177}$) or conversion of LK$^{220}$ to AA (LK$^{220}$/AA) disrupted the binding of TM9SF4 NTF to F-actin (Fig. 3c, d), as indicated by disappearance of respective fragments in pellet fraction (bound to F-actin), further confirming LK$^{220}$XXEX as the actin binding site.

The N-terminal region of TM9SF4 contains a putative copper redox center which exhibits NADH oxidase activity[18]. An in vitro NADH consumption assay was then conducted to measure the redox activity of TM9SF4 NTF. The results showed that GST-TM9SF4_N$^{24-258}$ and GST-TM9SF4_N$^{24-177}$ have NADH oxidase activity, while GST-TM9SF4_N$^{178-258}$ does not, suggesting that the redox domain locates within aa24–177 (Fig. 3e). We next investigated whether the redox activity of TM9SF4 NTF is copper-dependent. The results showed that the copper chelator bathocuproine alone failed to inhibit the activity of NADH oxidase (Fig. 3f). However, the NADH oxidase activity was lost when TM9SF4 NTF was unfolded by trifluoracetic acid and treated with the copper chelator at the same time and then refolded by buffer exchanging (Fig. 3f). Unfolding and refolding without copper chelator did not affect the NADH oxidase activity (Fig. 3f). These data confirmed that TM9SF4 NTF is a copper-dependent redox protein.

**TM9SF4 NTF induces actin oxidation**
Copper has been reported to oxidize actin via the formation of a cross-linked disulfide bond between Cys374 of actin monomers[23]. Therefore, we examined whether TM9SF4 NTF could induce G-actin oxidation. The results showed that incubation of G-actin with TM9SF4 NTF under non-reducing conditions promoted the formation of actin dimer (Fig. 3g), the effect of which could be prevented by N-ethylmaleimide (NEM) (Fig. 3h), a reagent that specifically blocks the Cys374 of actin[24]. These data suggest that TM9SF4 NTF can promote disulfide bond formation between Cys374 of actin monomers.

Actin is known to be a major target of S-glutathionylation in which actin is oxidized and a disulfide bond is formed between actin and glutathione (GSH), a tripeptide which functions as an important antioxidant[25,26]. Therefore, we next investigated whether TM9SF4 NTF could induce F-actin oxidation by measuring the S-glutathionylation level of actin. Here F-actin was incubated with GSH under non-reducing conditions, with or without TM9SF4 NTF, followed by immunoblot detection with GSH antibodies. We found that TM9SF4 NTF promoted F-actin glutathionylation, the action of which could be prevented by either NEM or a reducing agent dithiothreitol (DTT) (Fig. 3i).

Oxidation of F-actin may cause F-actin structural alteration, making F-actin more susceptible to cofilin-mediated severing[9]. Therefore, we examined whether TM9SF4 NTF could induce such a change of F-actin by a well-documented subtilisin digestion assay. In brief, subtilisin can cleave the peptide bond between Met47 and Gly48 to produce a 35 kDa fragment, while the oxidation of F-actin causes resistance to the proteolysis process[25]. Our results showed that incubation of F-actin with TM9SF4 NTF markedly decreased the subtilisin digestion of F-actin, further confirming that TM9SF4 NTF causes oxidation of F-actin (Fig. 3j).

**TM9SF4 NTF promotes cofilin-mediated actin disassembly**
To investigate how TM9SF4 NTF causes the depolymerization of F-actin, an in vitro biochemical actin depolymerization assay was performed by monitoring the shift of actin from the pellet (F-actin) to the supernatant fraction (G-actin). The results showed that TM9SF4 NTF alone had only a minor effect on F-actin depolymerization. However, TM9SF4 NTF, even at low concentration, could greatly enhance cofilin-mediated F-actin depolymerization (Fig. 4a), the effect of which was abolished by NEM treatment (Fig. 4b). We also used another fluorescent-based pyrene-actin depolymerization assay. The results confirmed that TM9SF4 NTF could enhance the cofilin-mediated F-actin disassembly, which could be prevented by NEM treatment (Fig. 4c). In addition, to eliminate the possibility that NEM treatment alone could affect cofilin-mediated actin depolymerization. F-actin (2.5 μM) were incubated with cofilin (1 μM) with or without NEM (2 mM), followed by measurement of fluorescent intensity change of pyrene-labeled F-actin. The result showed that NEM itself had no effect on cofilin-mediated-actin depolymerization (Fig. S14). To confirm the involvement of oxidation of actin Cys374 in TM9SF4 NTF-induced actin depolymerization, Cys374 was replaced with Ala by site-directed mutagenesis. The results showed that the binding affinity between actin and TM9SF4 NTF was not affected by actin mutation at Cys374. However, actin mutation at Cys374 nearly abolished the TM9SF4-facilitated actin oxidation/glutathionylation and F-actin disassembly (Fig. S15–16).

To verify whether cofilin is required for TM9SF4 NTF-induced actin depolymerization in cells, we generated stable cofilin knockdown NIH3T3 cells using lentivirus-based cofilin-shRNA (Fig. 4d). In control cells, overexpression of TM9SF4 NTF caused a dramatic change in cell morphology and the actin cytoskeleton (Fig. 4e). In contrast, over-expression of TM9SF4 NTF failed to disrupt the actin cytoskeleton in cofilin knockdown cells (Fig. 4e). Interestingly, an increased colocalization of TM9SF4 NTF and F-actin was found in cofilin knockdown cells, indicating that TM9SF4 NTF's binding to F-actin failed to induce

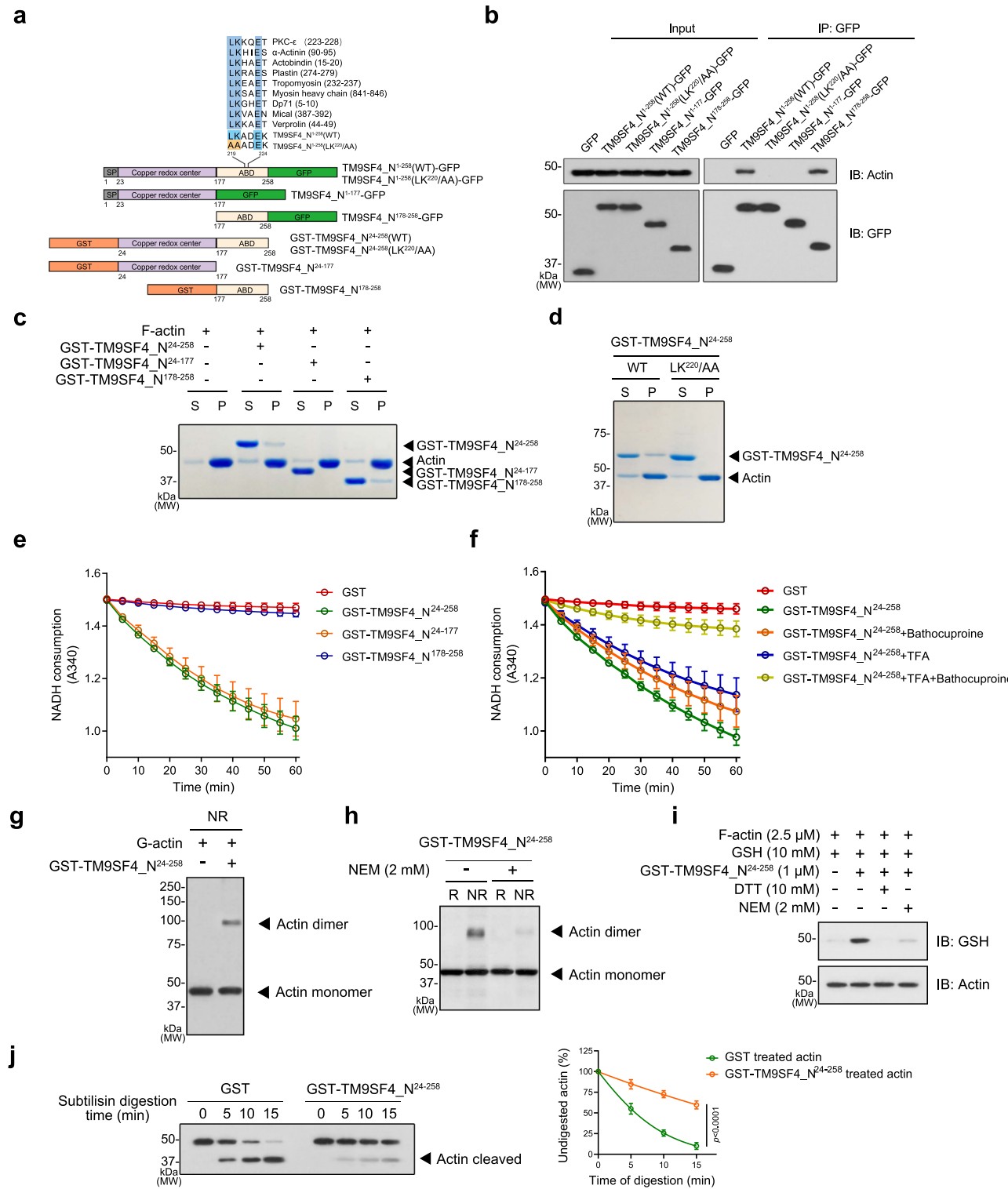

F-actin disassembly in the absence of cofilin. These data suggest that cofilin is essential for TM9SF4 NTF-mediated actin disassembly.

## TM9SF4 NTF cooperates with cofilin to regulate cell spreading

It has been reported that actin oxidation/glutathionylation level is elevated during cell spreading to regulate cell adhesion and locomotion[24]. To determine if TM9SF4 plays a role in cell spreading, the interaction among TM9SF4, actin and cofilin during cell adhesion was investigated in A2780 cells stably expressing GFP-TM9SF4. The subcellular localization results showed that, 1 h after cell seeding,

TM9SF4 was colocalized with both actin and cofilin at cell periphery and in some filopodia-like protruding structures (Fig. 4f). The colocalization almost disappeared at 4 h, and vanished at 24 h when the cells completely adhered to the substrate (Fig. 4f). We also observed that, 1 h after cell seeding, the protein glutathionylation signal was colocalized with F-actin at the cell periphery. In contrast, such a colocalization disappeared in TM9SF4 knockdown cells (Fig. 4g).

Next, co-immunoprecipitation experiments were performed to further study the interaction of TM9SF4 with actin and cofilin. The results also demonstrated that the interaction of TM9SF4 with actin

**Fig. 3 | TM9SF4 NTF directly binds to actin to induce actin oxidation.**
**a** Schematic representation of TM9SF4 NTF constructs. TM9SF4 NTF contains a putative copper binding center (purple) and an actin binding domain (ABD, pale yellow). **b** Co-immunoprecipitation analysis of the binding activity in the region of ABD. NIH3T3 cells were transfected with different constructs, followed by immunoprecipitation using GFP-Trap and analyzed by immunoblotting with anti-actin and anti-GFP antibodies. TM9SF4_$N^{1–177}$-GFP and TM9SF4_$N^{1–258}$ (LK$^{220}$/AA)-GFP failed to immunoprecipitate the actin. **c** Different GST-tagged TM9SF4 NTFs (2 µM) were incubated with F-actin (2.5 µM), followed by co-sedimentation analysis. S stands for the supernatant fraction, and P stands for pellet fraction. The co-sedimented proteins can be observed in the pellet fraction. **d** Actin co-sedimentation assay showing that GST-TM9SF4_$N^{24–258}$(LK$^{220}$/AA) failed to co-sediment with F-actin. **e, f** NADH consumption as measured by the decrease of absorbency at 340 nm. In **e**, different GST-tagged TM9SF4 NTFs (2 µM) were incubated with NADH (400 µM), followed by optical measurement. In **f**, GST-

TM9SF4_$N^{24–258}$ was treated with or without bathocuproine (25 µM), with or without unfolding by TFA and refolding. **g, h** G-actin dimer formation after incubation of G-actin (4.2 µM) with GST-TM9SF4_$N^{24–258}$ (2.5 µM) for 1 h. The samples were treated with or without 5% 2-ME as reducing (R) and non-reducing conditions (NR). *N*-ethylmaleimide (NEM, 2 mM) was used to block Cys374 of actin. **i** F-actin glutathionylation after incubation of F-actin (2.5 µM) with GST-TM9SF4_$N^{24–258}$ (1 µM) in the presence of 10 mM GSH. The samples were resolved in SDS-PAGE and immunoblotted with anti-GSH antibody. **j** Subtilisin digestion assay showing that GST-TM9SF4_$N^{24–258}$ treatment reduced the subtilisin digestion of F-actin. Shown are representative immunoblot images detected with anti-actin antibody (left) and summary data (right). Arrow head, cleaved 35 kDa actin fragment. Data are presented as mean ± SEM from 3 biologically independent experiments and two-tailed unpaired Student's *t*-test was used for statistical analysis. Source data are provided as a Source Data file.

and cofilin peaked at 1 h after cell seeding and decreased afterward (Fig. 4h). In agreement with the interaction pattern, the level of actin S-glutathionylation also peaked at 1 h after cell seeding (Fig. 4h). When treated with NEM, the interaction between cofilin and TM9SF4 was inhibited, while the interaction between actin and TM9SF4 was not affected, supporting the notion that TM9SF4 promotes cofilin-mediated actin depolymerization (Fig. 4h).

Together, these data agree with the role of cofilin-mediated actin depolymerization at the cell periphery during cell adhesion and cell spreading. It also supports a role of F-actin oxidation/glutathionylation in this process.

## TM9SF4 suppression alters cancer cell motility behavior, and reduces migration and invasion

A previous study has demonstrated that cofilin is involved in cancer cell motility and metastasis behavior[27]. Since TM9SF4 modulates cofilin-mediated actin cytoskeleton arrangements, we speculated that TM9SF4 may also affect cell motility in a similar manner to cofilin. Focal adhesion complexes link the actin cytoskeleton to the extracellular matrix, serving as anchorage devices that enable cell attachment to the extracellular matrix. When activated by numerous stimuli, such as growth factors, focal adhesion complexes function as a biosensor or integrator to control cell motility[28]. In our study, knockdown of TM9SF4 drastically increased the number and density of focal adhesion complexes, based on immunodetection of paxillin and vinculin, markers for focal adhesion complexes (Fig. 5a, b and Fig. S17).

ERM proteins (Ezrin, Radixin, Moesin) are known to provide a physical connection between actin cytoskeleton and plasma membrane, thereby participating in cell migration and cancer metastasis[29–32]. We also detected colocalization of TM9SF4 with Ezrin and actin stress fibers at the cell periphery (Fig. S18).

A time-lapse microscopy was used to monitor the cell motility. The cell center and cell boundary at each time point were traced into a consecutive line according to the timeline. The results showed that TM9SF4 knockdown cells followed a more linear path whereas the control cells moved around in a random direction (Fig. 5c). Analysis showed that knockdown of TM9SF4 increased the directionality of cell migration, but decreased cell turning frequency (Fig. 5d), indicating that knockdown of TM9SF4 changed the cell movement patten from random walking behavior to directional motility behavior.

The migration and invasion abilities of control and TM9SF4 knockdown ovarian cancer lines, including A2780, SKOV3 and several high grade serous ovarian cancer lines (HGSOCs; CaOV3, OVCAR3 and COV362), were also investigated by the transwell assay. Knockdown of TM9SF4 was found to significantly reduce the migration and invasion of A2780, CaOV3, OVCAR3, COV362 and SKOV3 cells, as indicated by reduced cell movement across transwell membrane within 24 h (Fig. 5e, f and Fig. S19). Consistently, we also observed that NIH3T3

cells overexpressing TM9SF4-NTF have increased migration and invasion properties (Fig. S20).

We also examined the role of TM9SF4 in colony formation and sphere formation. Individual tumor cells were seeded into solid culture of soft agar or suspension culture with stem cell medium, followed by determination of colony formation and sphere formation. The results demonstrated that TM9SF4 knockdown dramatically reduced colony formation and sphere formation (Fig. 5g, h and Fig. S21).

Taken together, these data suggest that TM9SF4 is involved in cancer cell motility and migration.

## TM9SF4 silencing suppresses tumor initiation and metastasis in vivo

To examine the role of TM9SF4 in tumorigenesis in vivo, human ovarian cancer cell lines A2780 and CaOV3 with or without TM9SF4 knockdown were subcutaneously inoculated in athymic nude mice (BALB/cAnNCrl-nu/nu). Rapidly growing tumors were found in mice injected with A2780 and CaOV3 cells expressing scrambled shRNA. In sharp contrast, we could rarely observe any tumor formation in mice inoculated with TM9SF4 knockdown cells (Fig. 6a, b and Fig. S22a).

We also examined the effect of TM9SF4 knockdown on ovarian tumor metastasis using the models established by others[33]. A2780 and CaOV3 cancer cells with or without TM9SF4 knockdown were injected intraperitoneally followed by metastasis analysis in peritoneal cavity (Fig. 6c, d and Fig. S22b). A2780 cells with or without TM9SF4 knockdown were also injected intravenously followed by metastasis analysis in lung (Fig. 6e–g). As expected, injection of control A2780 and CaOV3 cells resulted in numerous metastatic tumor nodules at metastatic sites (Fig. 6c–f and Fig. S22b). Strikingly, no metastatic tumor nodules could be observed following the injection of TM9SF4 knockdown cells (Fig. 6c–f and Fig. S22b). In agreement, hematoxylin-eosin staining confirmed the formation of metastatic nodules in lung tissues of the mice with A2780 injection, but not in mice injected with TM9SF4 knockdown cells. Furthermore, immunostaining with TM9SF4 antibodies demonstrated that TM9SF4 expression was much higher in tumor than that in normal tissue (Fig. 6g).

## Discussion

In the present study, we provide comprehensive evidence for the critical role of TM9SF4 in regulating F-actin dynamics and controlling tumor progression and metastasis. These include: (i) Knockdown of TM9SF4 in different types of tumor cells, which have a relatively high expression of endogenous TM9SF4, promoted actin stress fiber formation and increased the F-actin/G-actin ratio. (ii) An NTF of TM9SF4 could be found in cytosol of A2780 cells and binds to actin. (iii) In vitro analysis identified an actin binding domain and a redox domain in the TM9SF4 NTF. (iv) TM9SF4 NTF directly binds to F-actin to induce actin oxidation at actin Cys374, subsequently enhancing the cofilin-mediated F-actin disassembly, the effect of which is abolished by

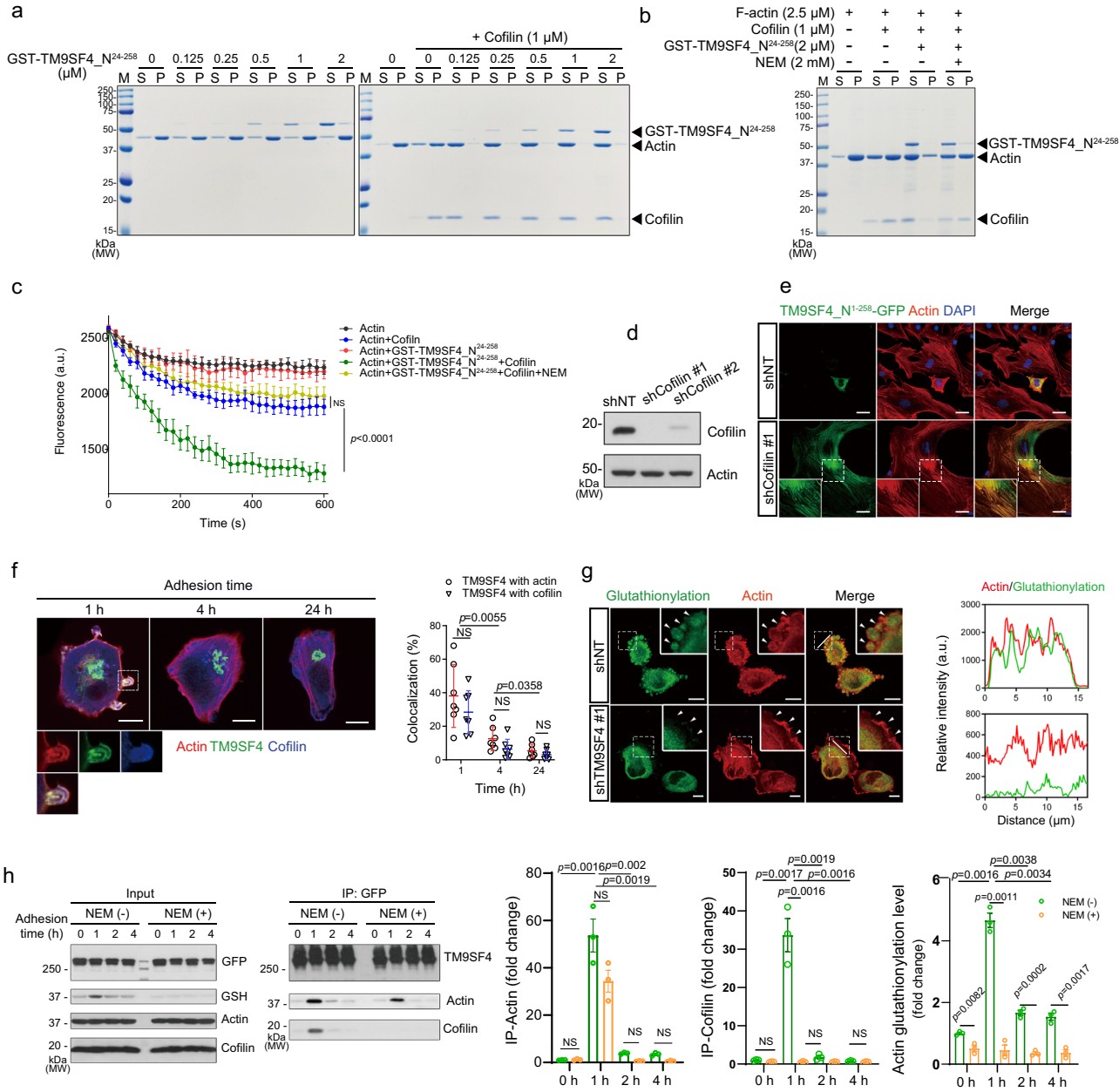

**Fig. 4 | TM9SF4 NTF promotes cofilin-mediated actin disassembly. a** GST-TM9SF4_N[24–258] promoted cofilin-mediated F-actin disassembly, as demonstrated by the shift of actin from pellet fraction (P, F-actin) to supernatant fraction (S, G-actin). **b** NEM treatment inhibited the action of GST-TM9SF4_N[24–258] in promoting cofilin-mediated actin disassembly. **c** Pyrene-actin depolymerization assay showing that GST-TM9SF4_N[24–258] accelerated cofilin-mediated actin disassembly, which was prevented by 2 mM NEM. **d**, **e** Cofilin is required for TM9SF4 NTF-induced F-actin disruption in NIH3T3 cells. The cells were stably transduced with lentivirus-based cofilin-specific shRNA (shCofilin #1), which could effectively knockdown the cofilin expression as in immunoblots (**d**). Scrambled shRNA (shNT) was used as control. In **e**, transient overexpression of TM9SF4_N[1–258]-GFP disrupted the F-actin fibers in control cells as monitored by Rhodamine Phalloidin staining, but failed to do so in cells with cofilin knockdown. Scale bar = 10 μm. **f** Time-dependent colocalization of TM9SF4, actin and cofilin at cell peripheral region of GFP-TM9SF4 overexpressing A2780 cells. Actin was monitored by Rhodamine Phalloidin staining, and cofilin

monitored by anti-cofilin antibody. Scale bar = 5 μm. **g** Colocalization of actin and S-glutathionylation signal in the cell periphery area of A2780 cells but not in cells with TM9SF4 knockdown. The cells were stained with anti-GSH antibody and rhodamine phalloidin 1 hr after cell seeding. Scale bar = 5 μm. Right panels showed the colocation along the white lines in the merge images in the left panel. **h** Time-dependent interaction of GFP-TM9SF4 with actin and cofilin as shown in representative co-immunoprecipitation images and summary data. A2780 cells were stably overexpressed with GFP-tagged TM9SF4, collected at different time points at 0, 1, 2, 4 h after seeding with or without 5 mM NEM pre-treatment, immunoprecipitated with anti-GFP antibody, immunoblotted by anti-actin and anti-cofilin antibodies. S-glutathionylated actin was detected by anti-GSH antibody in input. Data are presented as mean ± SEM from three biologically independent experiments and two-tailed unpaired Student's t-test was used for statistical analysis (**c**, **f**, **h**). Source data are provided as a Source Data file.

mutation at Cys374. (v) Knockdown of TM9SF4 increased cell adhesion and altered cell motility behavior of ovarian cancer cells A2780, SKOV3 and several HGSOCs. (vi) Knockdown of TM9SF4 nearly abolished the ovarian cancer growth and metastasis in athymic nude mice. Taken together, this study identified TM9SF4 as a protein that can promote

the cofilin-mediated F-actin disassembly. TM9SF4 may interact with cofilin to regulate cancer cell migration and metastasis. A schematic figure is shown in Fig. 7.

It is well documented that TM9SF4 plays important functional roles in phagocytosis of *Drosophila* hemocytes and cancer cell

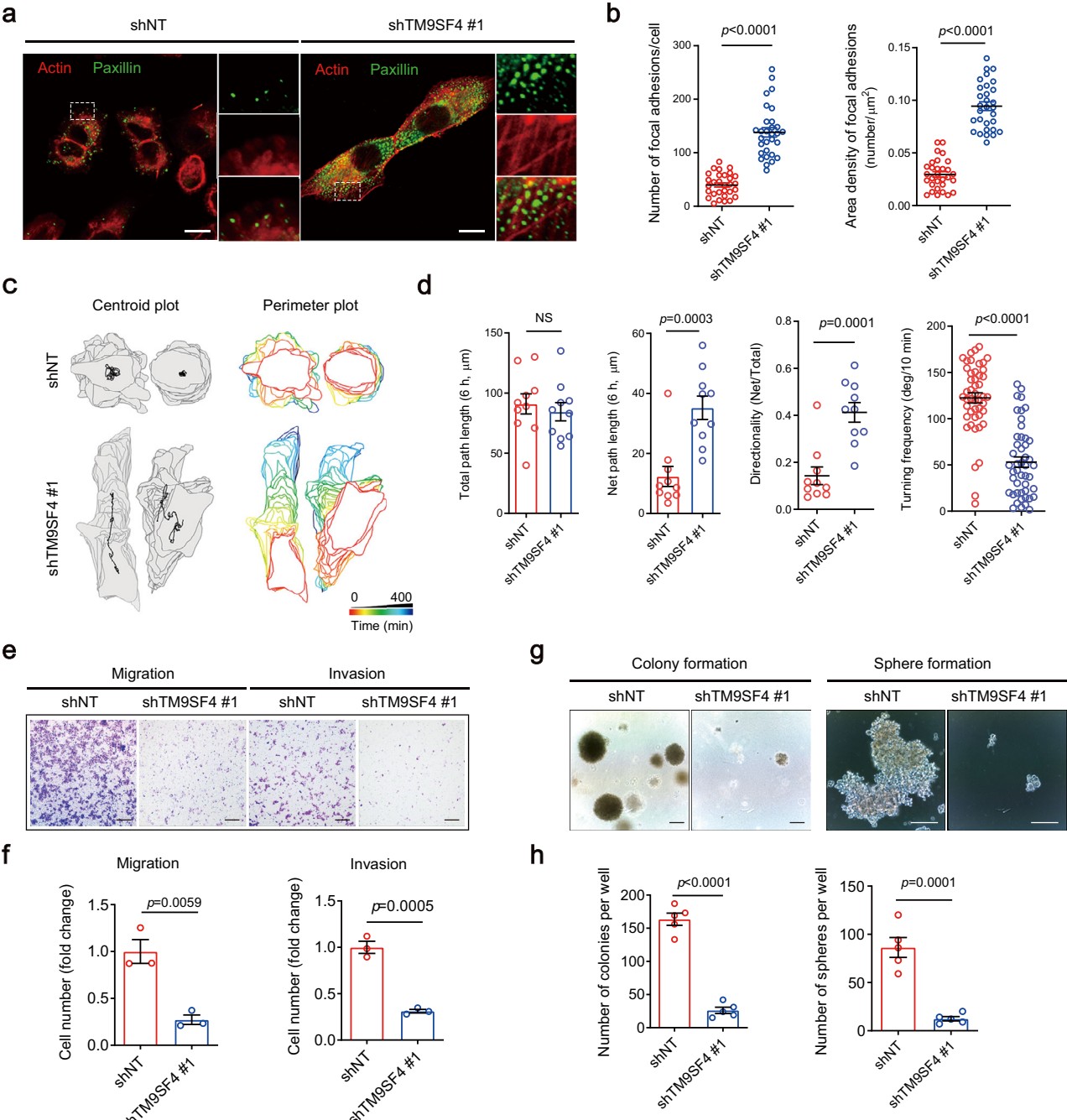

**Fig. 5 | Knockdown of TM9SF4 alters cancer cell motility behavior and reduces colony/sphere formation of A2780 cells. a, b** TM9SF4 knockdown increased the number and density of focal adhesion complexes in A2780 cells, as detected by anti-paxillin antibody. Shown are representative images (**a**) and summary data (**b**) (*n* = 5 biologically independent experiments, six fields of view per experiment). Scale bar = 10 μm. **c, d** TM9SF4 knockdown altered the motility behavior of A2780 cells. Shown are time lapse tracing of representative control cells and TM9SF4 knockdown cells (**c**) and summary data of cell motility parameters (**d**). Centroid plots showed the moving path of cell center over 400 min, depicted as black dot. Perimeter plots showed a stack of cell shapes as they move over time. In **d**, total path length is the total distance cells moved over a period of time; net path length is the net distance measured from the first cell position to the last cell position (or measured from the initial position of the cell to the final position of the cell); directionality is the ratio of the net path length to the total path length; turning

frequency is the average degree of turning every 10 min (*n* = 3 biologically independent experiments, 3–4 fields of view per experiment). **e, f** TM9SF4 knockdown inhibited the migration and invasion of A2780 cells. Shown are representative images (**e**) of migration and invasion assays and summary data (**f**) (*n* = 3 biologically independent experiments). Migration examined the number of cells that transversed the transwell membrane in 24 h. Invasion measured the cells transversed the transwell membrane pre-coated ;with Matrigel in 24 h. Scale bar = 200 μm. **g, h** TM9SF4 knockdown reduced colony formation in soft agar and sphere formation in serum-free stem cell medium with representative images (**g**) and summary data (**h**) (*n* = 5 biologically independent experiments). Scale bar = 100 μm. Data are presented as mean ± SEM and two-tailed unpaired Student's *t*-test was used for statistical analysis (**b, d, f, h**). NS, no significant difference. Source data are provided as a Source Data file.

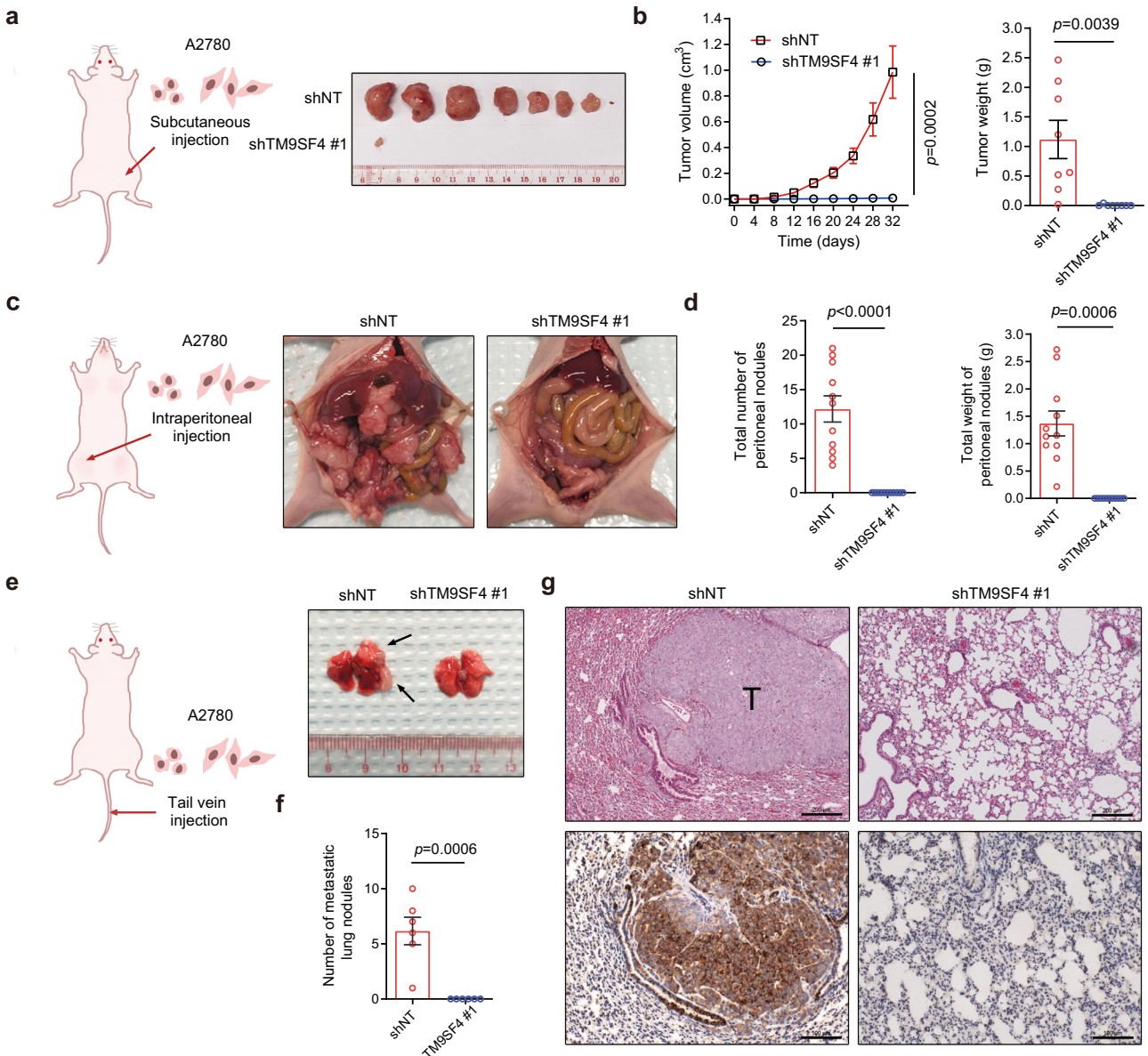

**Fig. 6 | TM9SF4 silencing suppresses tumor initiation/growth and metastasis by A2780 cells in athymic nude mice. a, b** TM9SF4 knockdown inhibited tumor initiation/growth. Control or TM9SF4 knockdown A2780 cells were injected subcutaneously into the hind leg of nude mice ($n = 8$ per group), followed by measurement of tumor sizes every 4 days. Shown are the dissected tumors. **c–g** TM9SF4 knockdown inhibited tumor metastasis to peritoneal cavity and lung. Control or TM9SF4 knockdown A2780 cells were injected intraperitoneally, followed by measurement of peritoneal tumor nodules in number and weight on day 30 (**c, d**) ($n = 11$ per group). Control or TM9SF4 knockdown A2780 cells were injected via ail vein, followed by measurement of lung tumor nodule number on day 32 (**e–g**) ($n = 6$ per group). Shown are representative images of peritoneal cavity with tumor nodules (**d**) and dissected lung tissue with tumor nodules (**e**), and summary data (**d, f**). **g** Representative images of H&E staining (upper row) and TM9SF4 immunoreactivity (lower row) of lung tissue sections from **e**. Scale bar = 200 μm (upper row) and 100 μm (lower row). T stands for metastasis tumor nodules. Data are presented as mean ± SEM and two-tailed unpaired Student's *t*-test were used for statistical analysis (**b, d, f**). Source data are provided as a Source Data file.

cannibalistic activity[11,13–16]. Phagocytosis is often referred to as macrophages engulfing unwanted material during immune defense. In free-living amoebae, phagocytosis ingests, kills and digests other microorganisms to feed upon them. Tumor cells can also feed upon neighboring cells to obtain nutrients, a process called cancer cannibalism, which is similar to that of amoebal behavior and immune cell phagocytosis[11,34]. A previous study showed that mutations in TM9SF4 altered cell morphology, resulted in an increased cell size and aberrant actin spikes in *Drosophila* macrophages[13]. However, the underlying mechanism of such TM9SF4-associated changes in cell morphology and behaviors is unknown. In the present study, we found that TM9SF4 expression was inversely correlated with actin

stress fiber formation. In cells that have relatively high expression levels of TM9SF4, such as A2780, MCF-7 and several HGSOCs, knockdown of TM9SF4 promoted actin stress fiber formation and increased cell size, whereas in cells that have very low expression levels of TM9SF4, such as NIH3T3 cells, overexpression of TM9SF4 NTF caused F-actin disassembly. Furthermore, in vitro studies showed that TM9SF4 NTF could directly bind to F-actin to promote the cofilin-mediated F-actin disassembly. These data clearly demonstrated that TM9SF4 can regulate actin dynamics, which provides a mechanistic explanation for previously reported TM9SF4-associated changes in cell morphology, phagocytosis, and cannibalistic activity.

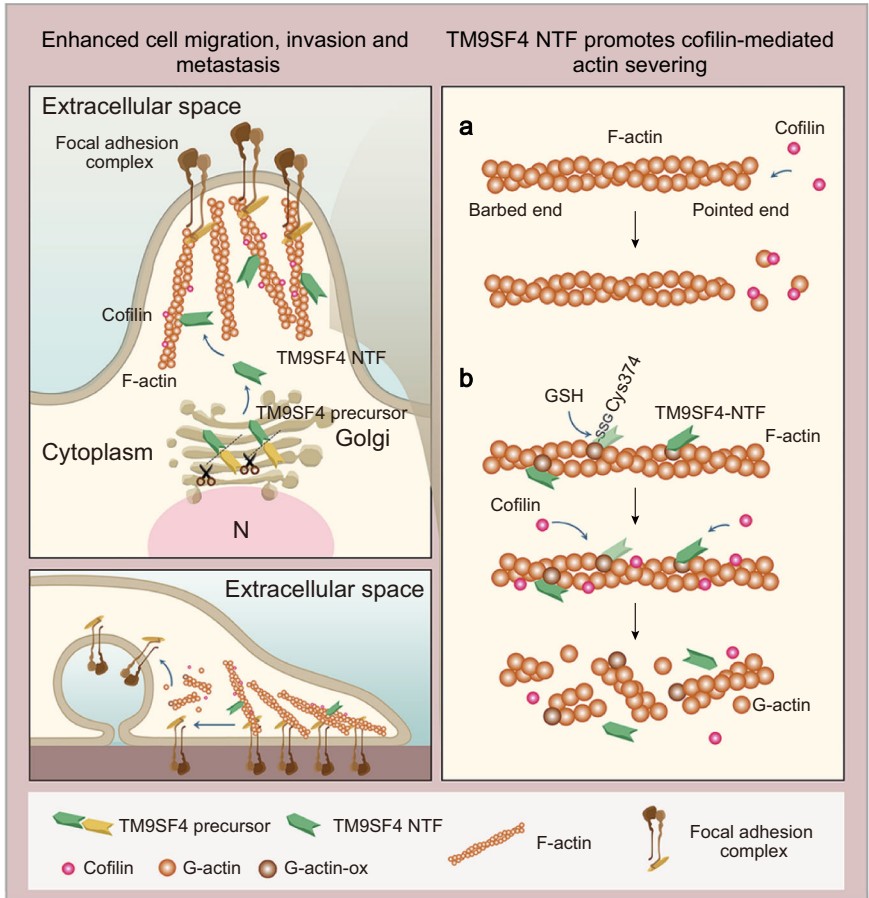

**Fig. 7 | A schematic model for actin disassembly via TM9SF4.** A schematic model is proposed based on the present findings. Upper left, top view; lower left, side view; right, schematics. **a** typically, cofilin can bind to F-actin at the pointed end and sever actin/ADP monomers from the end. This process is relatively slow and low-efficient. **b** TM9SF4 is cleaved to release its N-terminal fragment (NTF) into cytosol. The TM9SF4 NTF contains an actin binding site and redox domain. It binds to F-actin, induces actin oxidation and change the stability of actin structure. Such structural change enhances cofilin's severing activity to cause a rapid actin depolymerization. This TM9SF4-mediated actin disassembly affects cell adhesion and cell motility.

Actin dynamics promotes cell motility and cell invasion[35]. Cell motility requires dynamic rearrangement of the actin cytoskeleton as well as the coordinated assembly and disassembly of focal adhesions[36]. A balance in the turnover of focal adhesions is essential for cell motility[37]. Too little focal adhesion cannot provide sufficient traction force; whereas too much focal adhesion causes too strong cell attachment, which also hinders cell motility. In our study, TM9SF4 regulates actin dynamics, which is expected to alter cell motility. Indeed, knockdown of TM9SF4 was found to change the cell movement patten from random walking behavior to directional motility behavior in A2780 cells, and reduce the transwell migration and invasion of multiple ovarian cancer cells. Furthermore, TM9SF4 knockdown cells displayed an increased cell focal adhesion, which is consistent with its decreased motility. Note that some metastatic tumor cells show characteristics of apolar high turning frequency type of migration, which is positively correlated with metastatic potential[27]. In our study, metastatic ovarian cancer A2780 cells also displayed similar apolar high turning frequency type of motility. Knockdown of TM9SF4 reduced turning frequency of the cells, which agreed with its reduced metastatic ability[27].

ERM proteins (Ezrin, Radixin, Moesin) are well known to be involved in cell migration, invasion and metastasis[29,30]. Mechanistically, ERM proteins provide a physical connection between actin cytoskeleton and plasma membrane, thereby participating in cell migration and cancer metastasis[29,30]. Because TM9SF4 can regulate F-actin dynamics, it is conceivable that TM9SF4 may influence actin-to-ERM

connection, and thereafter regulate cell migration and cancer metastasis. Indeed, we did find some colocalization of TM9SF4 with ezrin and actin stress fibers at the cell periphery. However, further studies are needed to resolve whether TM9SF4 can indeed influence the actin-to-ERM connection.

Intriguingly, we found that the full-length TM9SF4 can be cleaved to form a TM9SF4 NTF, which resides in the cytoplasm of A2780 cells. This is consistent with results from several previous reports[18,38,39]. For example, one study detected a ~30 kDa cleaved form of TM9SF, which possesses the NADH oxidase activity, in human body fluid[18]. In another study, a 24 kDa N-terminal product was found to be cleaved from the yeast homolog of TM9SF4, named EMP70. This cleavage seems to depend on the Kex2p protein localized in Golgi lumen[38]. In yet another study, a 70 kDa TM9SF3, also known as hSMBP, was found to be cleaved into an active 34 kDa N-terminal polypeptide[39]. Therefore, cleavage of the hydrophilic N-terminus to generate a functional NTF could be a common feature for TM9SF family of proteins. This cleavage could be cell type-dependent, as we could find the cleavage in A2780 cells (Fig. 2) but could not detect it in NIH3T3 cells (Fig. S12).

Previously, two classes of oxidases, NOX and Mical, have been reported to regulate F-actin dynamics through actin oxidation[40,41]. The TM9SF family of proteins is reported to have NADH oxidase activity at their NTF[18]. Thus, we asked whether N-terminal oxidase of TM9SF4 plays a crucial functional role in F-actin disassembly. The results showed that TM9SF4 NTF was capable of inducing actin oxidation via forming a disulfide bond between actin monomers or between actin

and GSH, the latter one also being known as actin glutathionylation. Furthermore, the action of TM9SF4 NTF in promoting actin oxidation could be inhibited by DTT or NEM treatment. Cys374 is a highly reactive cysteine in actin that has been identified as an important site of actin oxidation[24,26,42]. Cys374 is also known to be crucial for the maintenance of F-actin stability. Therefore, TM9SF4-induced F-actin oxidation on Cys374 may weaken the stability of actin filament structure, making it vulnerable to cofilin-mediated mechanical severing; and thereafter shift the dynamic equilibrium between G-actin and F-actin towards depolymerization, resulting in disorganized actin filaments[43]. Indeed, we found that incubation of TM9SF4 with F-actin induced changes in F-actin conformation, as determined by altered susceptibility of F-actin to proteolysis. More importantly, TM9SF4-stimulated F-actin disassembly was abolished by prevention of F-actin oxidation with NEM (Fig. 4b, c) and by mutation at Cys374 (Fig. S16). These data strongly support the notion that TM9SF4 could regulate F-actin disassembly in a redox-dependent manner. Based on these evidences, we reasoned that a likely signaling cascade may include (i) TM9SF4 binding to F-actin to induce F-actin oxidation; (ii) Subsequent conformational change of F-actin to enhance the cofilin-induced F-actin severing and disassembly. Note that although our data showed a critical importance of TM9SF4 NTF in this action, we could not completely exclude the possibility that other part of TM9SF4 (membrane-associated part) could also have a role.

F-actin dynamics and cofilin regulation play an important role in cancer metastasis[1,2,44]. However, apart from a Mical-related report, it is poorly understood whether redox regulation of F-actin dynamics is critical for tumor growth and cancer metastasis[45]. Thus, we next explored possible role of TM9SF4 knockdown in ovarian cancer growth and metastasis in athymic nude mice. Strikingly, TM9SF4 knockdown nearly abolished the ovarian tumor initiation/growth and metastasis in all three models in athymic node mice. It is interesting to note that TM9SF4 knockdown phenocopies the loss of cofilin in many aspects, including the increased formation of stress fibers and focal adhesions, and altered migration behavior in metastatic cells[27,46]. Note that ovarian cancer, especially HGSOC, is notorious for its high metastasis and poor prognosis[47]. Therefore, the present study highlights an exciting possibility of targeting TM9SF4 or its related signaling pathway as a strategy for ovarian cancer treatment.

In conclusion, our present study uncovered an F-actin regulatory protein TM9SF4. TM9SF4 can bind to F-actin to induce actin oxidation, consequently enhancing cofilin-mediated F-actin disassembly. This mechanism plays a key role in migration and invasion of ovarian cancer cell lines. In athymic nude mice, knockdown of TM9SF4 completely abolished the ovarian cancer growth and metastasis, highlighting TM9SF4 as an attractive molecular target for ovarian cancer therapy.

## Methods

### Cell culture
Human embryonic kidney cell line HEK293FT, human Ovarian carcinoma cell line A2780, human breast cancer cell line MCF-7, and mouse embryo fibroblast cell line NIH3T3, were maintained in Dulbecco's Modified Eagle Medium (DMEM) supplemented with 10% FBS. Human breast cancer cell line T47D was cultured in RPMI 1640 Medium supplemented with 10% FBS. Human breast non-tumorigenic epithelial cell line MCF10A was grown in Dulbecco's Modified Eagle Medium/ Nutrient Mixture F-12 supplemented with 5% House Serum, 20 ng/mL recombinant human EGF, 10 μg/mL insulin, 0.5 mg/mL hydrocortisone, and 100 ng/mL cholera toxin. Unless otherwise stated, all culture media were supplemented with 50 U/mL penicillin G and 50 μg/mL streptomycin. All cells were kept at 37 °C in 5% $CO_2$.

### Immunoblotting
For immunoblotting analysis, cells were lysed in RIPA buffer supplemented with protease inhibitors cocktail (Roche). Protein concentration was determined using the detergent-compatible protein assay (Bio-Rad). Cell lysates containing equal amount of protein were subjected to SDS-PAGE. Proteins were then transferred to a polyvinylidene difluoride membrane (PVDF), and probed with appropriate primary antibodies overnight at 4 °C. The following antibodies were used in immunoblot analysis: TM9SF4 antibodies (1:2000, Proteintech), β-actin antibodies (1:2000, Santa Cruz), ERK antibodies (1:2000, Cell Signaling Technology), β-tubulin antibodies (1:1000, Santa Cruz), V5 antibodies (1:2000, Thermo Fisher), GFP antibodies (1:2000, TransGen Biotech), anti-GSH antibody (1:1000, Thermo Fisher) and cofilin antibodies (1:3000, Proteintech). Then, the membrane was incubated with HRP-conjugated secondary antibody (1:5000, Cell Signaling Technology) for 2 h at room temperature, followed by detection using ECL substrate (GE Healthcare). The images of protein bands were analyzed in a semi-quantitative manner through ImageJ 1.52p software. The uncropped scans of the blots were provided in the Source Data file.

### Immunostaining
Cells grown on coverslips were fixed in 4% paraformaldehyde, permeabilized with 0.1% Triton X-100, followed by incubation with primary antibodies overnight at 4 °C. The following primary antibodies were used: anti-GM130 antibody (1:1000, Proteintech), anti-GSH antibody (1:1000, Thermo Fisher), anti-paxillin antibody (1:500, Proteintech) and anti-ezrin antibody (1:250, NOVUS Bilogicals). The cells were then incubated with secondary antibodies (Alexa Fluor 488 donkey anti-mouse IgG/ Alexa Fluor 555 donkey anti-mouse IgG, 1:5000, Thermo Fisher) for 1 h at room temperature. DAPI (1:1000, Thermo Fisher) was used for nuclear staining. For F-actin staining, cells were fixed in 4% paraformaldehyde, blocked for 1 h and subjected to FITC-conjugated/rhodamine-conjugated phalloidin (1:1000, Thermo Fisher) staining at room temperature. The percentage of cells containing thick actin stress fiber with a diameter >0.5 μm was analyzed and used as an index for coarse quantification of the stress fiber. Immunofluorescence signal was detected using the Olympus FV1200 confocal system (FLUOVIEW Ver.4.2 software).

### Real-time quantitative PCR
Cells were harvested from 6-well culture plates and total RNA was extracted using RNAiso Plus reagent (Takara). Then, cDNA was generated using high-capacity cDNA reverse transcription kit (Thermo Fisher) and real-time quantitative PCR was performed using 7500 Fast Real-time PCR system (ABI) using Power SYBR Green PCR Master Mix (ABI). The primer sequences were as follows: TM9SF4 (human): 5'-GAT TGGTTGCCGTGGTCTTTA-3' (Forward), 5'-TTCTACGGGATCGTTCTGG TG-3' (Reverse); TM9SF4 (mouse): 5'-CTGGAGTCGCGCCAATCAAT-3' (Forward), 5'-GGCAGAAGGGCAATGAGTAGT-3' (Reverse).

### Cell viability assay
MTT (3-(4,5-dimethylthiazol-2-yl)-2,5-diphenyl tetrazolium bromide) was used to measure cell metabolic activity. Briefly, cells were seeded on the 96-well plate at a density of 3000/well. MTT prepared in PBS was added to a final concentration of 1 mg/mL. Cells were incubated with MTT for 4 h until crystalized formazan product is visible. DMSO (100 μL/well) was added and mixed by shaking several times to make an even distribution. The absorbance at 590 nm was recorded using spectrophotometer.

### Plasmids cloning
For transient expression of recombinant proteins, the full-length human cDNA of TM9SF4 was amplified by PCR and cloned into the pcDNA™6/V5-His A, pEGFP-N1, and pEGFP-C1 vectors for construction of the pcDNA6-TM9SF4-V5, pEGFP-TM9SF4-GFP, pEGFP-sp-mCherry-TM9SF4-GFP, and pEGFP-sp-GFP-TM9SF4 constructs. The SP sequence

for making the pEGFP-sp-mCherry-TM9SF4-GFP and sp-GFP-TM9SF4 constructs was based on the SignalP-5.0 Server prediction. TM9SF4_N$^{1-258}$-GFP, TM9SF4_N$^{1-177}$-GFP, and TM9SF4_N$^{178-258}$-GFP were constructed by replacing the full-length TM9SF4 in pEGFP-TM9SF4-GFP with the TM9SF4 aa1–258, aa1–177, and aa178–258, respectively. For stable expression of GFP-TM9SF4 proteins, GFP-TM9SF4 was subcloned from pEGFP-sp-GFP-TM9SF4 into a lentiviral vector pCDH-CMV-MCS. For recombinant protein purification, GST-TM9SF4_N$^{24-258}$, GST-TM9SF4_N$^{24-177}$, and GST-TM9SF4_N$^{178-258}$ were constructed by inserting TM9SF4 aa24–258, aa24–177 and aa178–258 into pGST-6p-1 vector, respectively. All mutant constructs were generated by PCR-based mutagenesis using Quickchange site-directed mutagenesis kit. All resulting plasmids were sequence verified.

### Lentiviral packaging and infection
Replication-deficient lentiviruses were produced using a 2nd generation lentiviral system. Briefly, lentiviral vectors were co-transfected with two helper plasmids (psPAX2, pMD2.G) into HEK293FT cells using a commercial lentivirus package kit (BioWit). The cell culture supernatant was collected at 48 h after transfection, centrifuged for 2 h at 50,000 × $g$, followed by re-suspension in cold PBS. Short hairpin RNA (shRNA) constructs were generated using pLKO.1 puro vector containing a cassette for puromycin. The shRNA target sequences used were: TM9SF4 (human) shRNA #1: GCGGATCACAGAAGACTACTA, TM9SF4 (human) shRNA #2: CTCGAACCCAGCTACCTTATG, TM9SF4 (mouse) shRNA #3: CTTACGAGTACTACTCATTG, TM9SF4 (mouse) shRNA #4: CGAGCGAATCACAGAAGAATA, cofilin (mouse) shRNA #1: GCAAGTCTTCAACACCAGAA, cofilin (mouse) shRNA #2: CAGA-CAAGGACTGCCGCTAT. Scrambled shRNA was used as the control.

### Subcellular fractionation
For separating cytoplasmic and membrane fractions, cells were harvested by a cell scraper in ice cold PBS, followed by sonication for several seconds to break the cells. Unbreaked cells were removed by centrifugation at 2000 × $g$ for 5 min at 4 °C. The supernatant was transferred to an ultracentrifuge tube and centrifuged at 15,000 × $g$ for 30 min at 4 °C. The supernatant was then collected as the cytoplasmic fraction. The pellet was resuspended in ice cold PBS containing 0.1% Triton X-100 and sonicated twice for 5 s each. The lysates were ultra-centrifuged again at 15,000 × $g$ for 30 min at 4 °C. The supernatant was collected as the membrane fraction. Samples were then analyzed by SDS-PAGE. For separating G-actin and F-actin in cells, A2780 cells treated with Jpk (50 nM) or TM9SF4 shRNA were lysed in LAS buffer (50 mM PIPES, pH 6.9; 50 mM NaCl; 5 mM MgCl2; 5 mM EGTA; 5% v/v glycerol; 0.1% NP-40, 0.1% Triton X-100, 0.1% Tween, 0.1% 2-mercaptoethanol, 1 mM ATP and protease inhibitor cocktail) and homogenized with a 25-gauge syringe. The cell lysate was centrifuged at 2000 × $g$ for 5 min to remove cell debris, and the supernatant was further centrifuged at 100,000 × $g$ for 1 h (Hitachi CS-150GXII Micro UITRA centrifuge, S140-AT Fixed Angle Rotor). The pellet fraction which contains the filament actin was resuspended with same amount of LAS buffer supplemented with 1% SDS. Samples were then analyzed by SDS-PAGE.

### Co-immunoprecipitation
For identifying the interacting partners of TM9SF4, A2780 and NIH3T3 cells, with or without GFP-TM9SF4 overexpression, were lysed in lysis buffer (50 mM Tris-HCl, pH 7.5; 150 Mm NaCl; 1% NP-40; 0.5% sodium deoxycholate; 0.2% Triton X-100 and protease inhibitor cocktail) at 4 °C. After centrifugation at 15,000 × $g$ for 10 min at 4 °C, supernatant was collected, and protein extract was incubated with TM9SF4 antibodies or GFP antibodies for 2 h at 4 °C under gentle rotation. Protein G-agarose was then added and incubated overnight at 4 °C under gentle rotation. The bound proteins were washed three times with washing buffer (50 mM Tris-HCl, pH 7.5; 500 mM NaCl; 0.1% NP-40; 0.05% sodium deoxycholate) and eluted in SDS sample buffer. Immunoprecipitates were subjected to SDS-PAGE and analyzed by immunoblotting. For co-immunoprecipitation in A2780 cells with stable expression of GFP-TM9SF4, whole-cell lysates were extracted with IP buffer (50 mM Tris-HCl, pH 7.5; 150 mM NaCl; 1% NP-40 and 5% glycerol and protease inhibitor cocktail). Extracted proteins were incubated with GFP antibodies or Actin antibodies, respectively, at 4 °C with rotation for 2 h. Protein G-agarose was then added and incubated overnight at 4 °C with rotation. The bound proteins were washed and eluted as above mentioned and then analyzed by immunoblotting.

### Mass spectrometric analysis
After electrophoresis, SDS-PAGE gels were stained with Coomassie blue solution for 30 min at room temperature. The protein bands of interest were then excised into 1 × 1 mm pieces and equilibrated in 200 μL of 50 mM NH$_4$HCO$_3$ twice for 20 min each, followed by addition of 50 mM NH$_4$HCO$_3$/ acetonitrile (1:1) and soaked for 20 min or longer until the gel chips lose color. The supernatant was replaced with acetonitrile to dehydrate the gel. After drying, the gels were rehydrated with 10–15 μl sequencing grade trypsin (10 ng/ml in 25 mM NH$_4$HCO$_3$ buffer, pH 8.0, Promega) and incubated overnight at 37 °C. Spectrometric analysis was conducted on an Ultraflex MALDI-TOF (Bruker Daltonics) using a Scout ion source and operating in a positive reflectron mode.

### Purification of TM9SF4 NTFs
E. coli strain BL21 (DE3) was transformed with pGST-TM9SF4_N$^{24-258}$, pGST-TM9SF4_N$^{24-258}$ (LK$^{220}$/AA), pGST-TM9SF4_N$^{24-177}$ and pGST-TM9SF4_N$^{178-258}$ plasmids, respectively. Protein expression was induced with 1 mM IPTG at OD600 = 0.8 and carried out overnight at 16 °C. Cells were harvested by centrifugation and then lysed by sonication in 20 mM phosphate buffer supplemented with 0.5 mg/mL lysozyme and protease inhibitors. The insoluble materials were removed by centrifugation at 21,000 × $g$ for 15 min at 4 °C. The column was filled with Glutathione Sepharose 4B beads after fully mixing and equilibrated using the GST column binding buffer (50 mM Tris-Cl, pH 8.0; 50 mM NaCl). The cell lysate was applied to the column and then washed with GST binding buffer several times until no protein containing in the flow through fraction. The recombinant protein was then eluted in GST elution buffer (50 mM Tris-Cl, pH 8.0; 50 mM NaCl; 10 mM GSH).

### F-actin cosedimentation assay
Actin was purchased from Cytoskeleton, Inc. and polymerized by addition of actin polymerization buffer (5 mM Tris-HCl, pH 8.0; 0.2 mM CaCl$_2$; 50 mM KCl; 2 mM MgCl$_2$ and 1 mM ATP) and incubated for 1 h at room temperature. Purified TM9SF4 NTFs were incubated with polymerized F-actin for 2 h at room temperature and then samples were ultracentrifuged in a TLA100 rotor (Beckman Instruments) at 150,000 × $g$ for 1.5 h at room temperature. Supernatants and pellets were separated and reconstituted in equal volume of SDS sample buffer. The pellet and supernatant fractions were analyzed by SDS-PAGE.

### Actin depolymerization assay
Pyrene-conjugated rabbit skeletal muscle actin (Cytoskeleton Inc.) was used to measure the polymerization state of actin according to manufacturer's instructions. Pyrene G-actin was resuspended to 1 mg/mL in G-actin buffer (5 mM Tris-HCl, pH 8.0; 0.2 mM CaCl$_2$; 0.2 mM ATP and 1 mM DTT) and polymerized into F-actin by adding 10× actin polymerization buffer (500 mM KCl, 20 mM MgCl$_2$ and 10 mM ATP in 100 mM Tris, pH 7.5) to a final concentration of 0.25×. F-actin (2.5 μM) were incubated with cofilin (1 μM) with or without GST-TM9SF4_N24-258 (2 μM), followed by measurement of fluorescent intensity of

pyrene-labeled F-actin every 20 s over 10 min. Fluorescence intensity was measured on a SpectraMax i3X at 407 nm with an excitation wavelength of 350 nm.

## NADH consumption assay
NADH absorbs light at 340 nm, while the oxidized NADH loses such ability. The redox activity of TM9SF4 NTF was determined by measuring the rate of NADH oxidation, which is recorded spectrophotometrically every 5 min for 60 min and quantitated using changes in absorbance at 340 nm in a reaction mixture containing 20 mM Tris-HCl pH 8.0, and 200 μM NADH at 37 °C with stirring. For unfolding and refolding, 50 μg recombinant TM9SF4_N258 in a total volume of 100 μL was mixed with 1 μL trifluoracetic acid. Bathocuproine (25 μM) was added to the mixture to remove the buried copper. After incubation for 2 h at room temperature, the buffer was changed with 20 mM Tris-HCl pH 8.0 using 10 K centrifugal filters to refold the proteins.

## Subtilisin cleavage assay
F-actin (2.5 μM) was incubated with GST-TM9SF4_N$^{24-258}$ (2 μM) or GST (2 μM, control) for 2 h at room temperature. Samples were then incubated overnight at 4 °C for complete actin depolymerization. Digestion by subtilisin was carried out with a subtilisin to actin mass ratio of 1:1000 at room temperature. The digestion process was stopped by adding 5× SDS loading buffer. The samples were analyzed on a 15% SDS-PAGE and detected by anti-actin antibody. Densitometry analysis was performed using ImageJ software.

## Histology
Tissue samples were deparaffinized and rehydrated. Heat-induced antigen retrieval was performed in a time-setting water bath. Slides were washed with TBS plus 0.025% Triton X-100 for 5 min twice with gentle agitation and blocked with 3% BSA in TBS for 1 h at room temperature. Then, samples were incubated with diluted primary antibody at 4 °C overnight, followed by incubation with HRP-conjugated secondary antibody for 1 h at room temperature. Hematoxylin counterstain was used to show cellular nucleus. The images were captured using the Q-Imaging digital Camera and Carl Zeiss Axiophot 2 Microscope Integrated Biological Imaging System.

## Time-lapse imaging
Control and TM9SF4 knockdown A2780 cells were seeded onto 35-mm glass-bottom dishes (SPL Life Sciences). After 24 h, cells were imaged using Nikon Ti-2 Inverted Fluorescence Microscope overlaid with motorized stage and 37 °C incubation System. Data collected from movies captured for 6–7 h (at a rate of 1 frame/5 min).

## Migration and invasion assays
Cell migration and invasion assay was performed by using 24-well Transwell (SPL, 8 μm pores) coated without or with Matrigel (BD Biosciences). Cells (1 × 10$^5$/mL, 200 μL) were transferred onto the upper chamber in serum-free condition and cultured for 24 h allowing the cells to move through the microporous membrane to the lower chamber containing 10% serum. The cells on the underside of the inserts were fixed with 4% paraformaldehyde for 30 min and stained with 0.1% crystal violet. Randomly selected fields on the fixed transwell chambers were counted and photographed.

## Colony formation and sphere formation assays
For colony formation assay, a 6-well plate was coated with 1% agar and 2× media at 1:1 ratio and left to solidify for 30 min. Top portion was prepared with 0.6% agar and 2× media with cells plated at a density of 5,000 cells/mL. Pictures were taken at day 10 using Nikon Ti-S inverted microscopy.

For sphere formation assay, 1000 cells were cultured in 6-well ultra-low attachment plates in serum-free medium supplemented with 20 ng/mL epidermal growth factor (EGF) (PeproTech), 10 ng/mL basic fibroblast growth factor (bFGF) (PeproTech), and B27 supplement (Thermo Fisher). Culture medium was replaced every 2 days.

## Mice
Female nude mice (BALB/cAnNCrl-nu/nu) of 4–6 weeks old were used. All animal experiments were approved by the Animal Experimentation Ethics Committee of the Chinese University of Hong Kong and performed in compliance with the Guide for the Care and Use of Laboratory Animals (National Institutes of Health publication, 8th edition, updated 2011). The maximal tumor burden permitted by our ethics committee was 20 mm at the largest diameter and tumor volume must not exceed 2000 mm$^3$.

## Tumor growth and metastasis measurements
A2780 cells stably expressing TM9SF4 shRNA or scramble shRNA were harvested using 0.05% trypsin, washed once with PBS, and counted prior to injection. For subcutaneous transplantation, 1 × 10$^7$ cells in 50 μL of PBS were injected subcutaneously into the hind leg of nude mice. Tumor dimensions were measured using a caliper every 4 days, volume was calculated by using formula: Volume (cm$^3$) = (width)$^2$ × length/2. Tumors were collected after 32 days and tumor weight was measured. The tumor volumes were measured every day. For tumor metastasis model, 5 × 10$^6$ cells in 150 μL of PBS were intraperitoneally injected into nude mice. Total number of peritoneal nodules was counted, and total weight of peritoneal nodules was measured after 30 days. We also monitor pulmonary nodules to measure tumor metastasis in mice intravenously injected with 1 × 10$^6$ cells in 100 μL of PBS. Number of metastatic lung nodules was counted after 30 days.

## Statistics
All data are presented as means ± SEM of at least three independent experiments. Fluorescent-imaging analysis and animal studies were performed blinded and randomized. Calculations were done using Microsoft Excel 2019 and graphs were plotted in GraphPad Prism 8 version 8.0.2. The sample size and P-values are mentioned in the figure graphs or the figure legends. Comparisons between two groups were measured by two-tailed unpaired Student's t-test. Differences with P-values < 0.05 were considered statistically significant.

## Data availability
The data supporting the findings of this study are available within the paper and its Supplementary Information files. Source data are provided with this paper.

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

## Acknowledgements

This work was supported by grants from Hong Kong Research Grant Committee (AoE/M-05/12 to L.J. and X.Y., GRF/14100619 to X.Y., RIF/R4005-18F to X.Y.), and Health and Medical Research Fund (06170176 to X.Y.).

## Author contributions

Z.M., Z.L., L.J., and X.Y. conceived and designed the experiments. Z.M. performed most of the experiments. X.Y. and L.J. provided funding support. Z.L. performed the molecular cloning. M.X. and H.Y. performed the construction of animal model. Z.M. and X.Y. wrote the manuscript. All authors approved the final version of the manuscript.

## Competing interests

The authors declare no competing interests.
