## [Peer Review File · Nature Communications]

TM9SF4 is an F-actin disassembly factor that promotes tumor progression and metastasisReviewers' comments:

Reviewer #1 (Remarks to the Author), Expertise: proton channels, cancer:

TM9SF4 is a novel F-actin disassembly factor that promotes 2 cancer cell migration and metastasis

By Zhaoyue Meng

The issue is of course of interest but the authors disregarded a number of important details in their experimental design:

1. Previous reports have already shown that TM9SF4 is involved in cancer cells motility and migration through knockdown experiments (Lozupone et al EMBO Reports 2009) and therefore what the authors show is not original, but the experiments on actin assembly.
2. Actually, actin function in cancer cells is entirely dependent on the actin to plasmamembrane connection through ERM (Fais S. Trends Mol Med. 2004; Brambilla D Int J Cancer. 2009 Federici C Int J Cancer. 2009). This can't be ignored and the authors should understand whether TM9SF4 may influence the actin-to-ERM connection. This appears highly conceivable inasmuch as TM9SF4 has been described to be related to the cannibal activity of tumor cells that in turn was related to ERM function as well (Lugini et al Cancer Res 2006). In particular this was described as an activity of tumor cells deriving from metastatic lesions and not from primary lesions. Did the authors had a chance to test it for the actin assembly as well?
3. The authors should comment all the data supporting the relationship between TM9SF4 and a pivotal activity of malignant cells such as cannibalism or cell-in-cell phenomena in general, as it has been commented in a recent perspective article (Fais S, Overholtzer M. Nat Rev Cancer. 2018). Of interest TM9SF4 has an homologous in amoebas and some analogies between cancer cells and amoebas have been proposed (Fais S, Fauvarque MO. Trends Mol Med. 2012)
4. A previous paper has shown that TM9SF4 is involved in the V-ATPase activity, in turn representing a key mechanism favouring the extracellular acidification of cancer cells (Lozupone et al Oncogene 2015). Actually, tumor cells live in acidic condition and many studies have shown that the same cell lines markedly change as cultured at different pH conditions (e.g. De Milito et al Int J Cancer 2010; Logozzi et al Cancers 2018; Urbanelli et al J Enzyme Inhib Med Chem. 2020). In trying to reproduce the in vivo condition the authors should perform new experiments comparing the various cell lines at least at the standard pH condition (7,4) and at the tumor pH (6,5). I guess it is worthy to be mentioned that tumors are acidic and that probably all the data obtained with tumor cells cultured at pH 7,4 are not reliable, paradoxically a buffered pH may cure tumors (Pillai SR, et al .Cancer Metastasis Rev. 2019)

The manuscript needs extensive english revision

Reviewer #2, Expertise: actin dynamics, cancer metastasis (Remarks to the Author):

In this manuscript by Meng et al. submitted to Nature Communications, the authors investigate the role of TM9SF4 in cell cancer cell behaviours and actin cytoskeleton dynamics. The authors provide evidences that knocking down TM9SF4 increases cell spreading, affect cell polarity and promotes actin filament assembly. These cellular defects are associated with a reduction in cancer cell abilities to migrate, invade and self-renew in vitro and to grow and to form metastasis in nude mice. Using a very elegant experiment, the authors also show that TM9SF4 is cleaved to produce a N-terminal fragment (NTF) with abilities to bind to F-actin to induce actin oxidation and to enhance cofilin-mediated F-actin disassembly. The authors propose that the cleaved NTF of TM9SF4 promotes cofilin-mediated F-actin disassembly to regulate cancer cell migration and metastasis. This work is interesting and the experimental data provided are globally well done. I have nevertheless a few concerns about some of the data, their interpretation data and the conclusion draw by the authors.

Majors points:

1. In some Western Blots, actin is used as loading control, as TM9SF4 controls actin dynamics, it could be wise to provide evidences that TM9SF4 does not affect total actin levels.
2. The authors claim that the number of actin stress fibres is increased in TM9SF4 knocked down cells (Fig. 1e). I could not find information on how stress fibres were evaluated but if they were

evaluated based on Phalloidin staining only, this might not be sufficient to make such a claim. If the authors are referring to actin fibres crossing the cells, they could be ventral stress fibres but they could also be others actin structures, like TAN lines. Are the actin fibres identified by the authors attached to Focal Adhesions, are phospho-myosin II positive and found basal to the nucleus? Related to this point is the localization of Paxillin as a marker of Focal Adhesion in Fig. 5a. It would be important to verify that other Focal Adhesions markers show the same accumulation in dot-like structures (Zyxin? Vinculin?) because based on the image provided for cells knocked down for TM9SF4, it does not look like that the Paxillin punctuated staining are associated with actin stress fibres. It could be trafficking defect of Paxillin, which accumulates in vesicles.

3. Conversely, the authors claim that overexpression of TM9SF41-258 caused a drastic disruption of F-actin fibres in NIH3T3 cells (Fig. 2h). From the picture provided, we can see that TM9SF41-258 induces a kind of clustering of F-actin, which could prevent the visualization of actin fibres. Does overexpressing the full length induce the same phenotype?
4. TM9SF4 knocked down cells appear very round in Fig. 1c but form many extruding spikes in Fig. 1f. It makes it hard to conclude on the role of TM9SF4 on cell shape? Are cells in Fig. 1c representative of the phenotype?
5. Could the authors discuss how it could be that apolar cells (control cells) migrate faster than polar cells (cells knocked down for TM9SF9) that appear to form more striking lamellipodia as shown in Fig. 1c or 4g?
6. In Extended data Fig. 1, the authors show that knocking down TM9SF4 reduces cell growth (the sum of proliferation and apoptotic events). In the text they claim that TM9SF4 promotes cell survival, while in the legend, they claim, it promotes proliferation. Are TM9SF4 knocked down cells die more or proliferate less? They also show that TM9SF4 knocked down cells are binucleated (Extended data Fig. 2). Could the decrease in migration, invasion and metastasis in nude mice due to the lack of cell survival or binucleated phenotypes? This is important to conclude on the direct or indirect effect of TM9SF4 on cell migration.
7. The GFP levels in mutation/depletion of the ABD is very weak and not concentrated in puncta (Extended data Fig.6) , compared to the TM9SF41-258 in Fig. 2h. Are these mutation/deletion versions unstable? If not, the intensity of GFP should be the same for the 4 TM9SF4-GFP constructs. This is important to determine whether they co-localize with F-actin or not.
8. Based on the model provided, it would be important to provide a piece of data showing that cells expressing TM9SF41-258 (NIH3T3 for example) migrate/invade more in vitro or grow more/bigger spheres.

Minor points:

1. Although the text could benefit from some corrections, it is in general well written. However, one part that could be improved substantially to make the manuscript more appealing is the discussion. This part is mostly a repetition of the result part, with little discussions in the context of the literature. This is revealed by the fact that the first half of the discussion does not contain any reference. The model in Fig. 7 could be discussed in the context of the current knowledge.
2. The authors used different cell lines to perform this study but it is not always clear which cell line have been used for each experiment. For example, in Fig. 1f, are those A2780 or MCF7 cells? This is important to compare data with other panels using these cells. These information could be easily included in the legends

Reviewer #3, Expertise: actin dynamics, biochemistry (Remarks to the Author):

The manuscript "TM9SF4 is a novel F-actin disassembly factor that promotes cancer cell migration and metastasis", by Meng and colleagues, addresses the N-terminal part of TM9SF4 regulates actin dynamics. In particular, it highlights that this N-terminal fragment possesses and synergize with cofilin to tune actin turnover in cells.

The paper is well written and a large array of experimental approaches allows to convincingly gain further insight into the activity of TM9SF4.

Though I have a major concern about the molecular description that is made in the discussion section of the manuscript on how the NTF of TM9SF4 may cooperate with cofilin to induce actin disassembly in vitro : There is no proof that any actin conformational change caused by TM9SF4_NTF that would enhance cofilin disassembly activity. As explained in details below, I think the biochemical conclusion is not scientifically correct and the discussion should be modified to reflect the exact knowledge gained by the authors in this manuscript. Alternatively, additional experiments to derive more advanced molecular details should be designed and performed. In

general, the method section for the in vitro characterization of TM9SF4 is not describing the experiments with enough details, especially buffer conditions and product concentrations. Major modifications should therefore be made before the manuscript can be considered as scientifically valid.

Major points:

- figure 2a : signal peptide color should be modified to match between what is shown in the sketch and the diagrams. Also, please indicate the TMDx domains, as well as the amino-acid positions in the sequence.
- line 234: I agree with the results shown figure 3j, that subtilisin cleavage is impaired in the presence of TM9SF4_NTF, but I disagree with the interpretation that TM9SF4_NTF induces an F-actin twist. First, in the method section, there is no indication whether the assay is performed in non-reducing condition or in the presence of GSH to induce glutathionylation. The binding of TM9SF4_NTF alone could protect actin from subtilisin digestion.
- line 243, figure 4b: The authors do not show that NEM treatment alone prevented the binding of cofilin, and therefore its depolymerizing activity. This point has been addressed in the past by others (for example, by the Emil Reisler lab: <https://doi.org/10.1021/bi400715z>). As a consequence, the conclusion derived from this assay should be reassessed.
- line 200 : it is unclear if the affirmation made in the sentence refers to the article in the reference 26 or to the results shown in figure 3a.
- line 890 : the method section lacks information about the buffer used for NADH consumption and NADH concentration.
- Are the authors able to exclude that oxidation is not happening on Met 44 and Met47 in the presence of NADPH ?
- figure 7: the cofilin has a weird shape.... please redraw it with its known shape and size, relative to actin subunits.
- figure 7: the data do not show that cofilin is specifically recruited onto oxidized actin subunits, as shown in the model shown in figure 7. Please avoid confusing description of the conclusion of the performed experiments in the model figure.
- line 384 : the sentence 'Lastly and more importantly, TM9SF4-stimulated F-actin disassembly was abolished by prevention of F-actin oxidation with DTT/NEM. ' is not backed up by any data. This is a very important point. Unless I misread the entire manuscript, this is never shown. For me, one possible explanation for TM9SF4_NTF being able to increase cofilin-induced F-actin disassembly is pure binding competition for actin residues (eg on Sub-domain 2 and/or sub-domain 1 where Cys374 is located). This binding competition is enough to explain data of figure 4 a, b and C. The cofilin-concentration dependence of F-actin disassembly is not straightforward (eg severing activity versus filament saturation by binding which prevents cofilin severing) and extra care should be made when trying to conclude from these types of assay. Furthermore, there is no indication in the method section that those assay were performed in non-reducing conditions or in the presence of NADPH or GSH, so the TM9SF4_NTF induced oxidation/glutathionylation might be minimal or even non-existent. This point should be discussed correctly or new experiments with the appropriate conditions should be performed to be able to derive the correct conditions. For example, the oxidation could be performed on F-actin by TM9SF4_NTF, the actin filament separated from TM9SF4_NTF by centrifugation before cofilin is added to induce depolymerization.

minor points:

- ref 34 and 37 are the same.
- typo line 166 GTP for GFP.
- line 179 : the statements on GFP and GST construct could be removed.
- figure 3 legend, especially for the subsets c,d and e are not correctly mentioned in the legend and more details should be given.
- line 862, I am guessing that the TM9SF4_N178-258 construct also contains a GST tag, like all the other constructs, right ?
- some actin reference literature are sometimes a bit outdated....
- line 347: 'it is clear that...' could be replaced with something less direct, like 'our data strongly indicates'.
- line 374, the reference to NADH consumption for oxidation is misplaced, as it would indicate that it refers to the activity of TM9SF family proteins.

Point-to-point response to reviewers

Reviewer #1 (Remarks to the Author), Expertise: proton channels, cancer:

TM9SF4 is a novel F-actin disassembly factor that promotes 2 cancer cell migration and metastasis

By Zhaoyue Meng

Comments from reviewer#1: The issue is of course of interest but the authors disregarded a number of important details in their experimental design:

1. Previous reports have already shown that TM9SF4 is involved in cancer cells motility and migration through knockdown experiments (Lozupone et al., EMBO Reports 2009) and therefore what the authors show is not original, but the experiments on actin assembly.

Answer: We thank the reviewer for the comment. Lozupone et al., reported that TM9SF4 affects cancer cell motility and migration (Lozupone et al., EMBO Reports 2009;10:1348-1354; Lozupone et al., Oncogene 2015;34:5163-5174). However, they did not investigate actin involvement. Our present study provides ample amount of evidence, uncovering a novel mechanism of how TM9SF4 can regulate cofilin-mediated actin disassembly, thereafter, controls cancer cell motility and migration. Therefore, our data not only confirmed Lozupone's results, but more importantly, provided a detailed mechanistic insight of how this can happen. The previous reports from Lozupone et al., were already cited and mentioned in Introduction.

2. Actually, actin function in cancer cells is entirely dependent on the actin to plasma membrane connection through ERM (Fais S. Trends Mol Med. 2004; Brambilla D Int J Cancer. 2009 Federici C Int J Cancer. 2009). This can't be ignored and the authors should understand whether TM9SF4 may influence the actin-to-ERM connection. This appears highly conceivable inasmuch as TM9SF4 has been described to be related to the cannibal activity of tumor cells that in turn was related to ERM function as well (Lugini et al Cancer Res 2006). In particular this was described as an activity of tumor cells deriving from metastatic lesions and not from primary lesions. Did the authors had a chance to test it for the actin assembly as well?

Answer: Thank you for your great input. We have taken your suggestion and examined colocalization of TM9SF4 with ezrin and F-actin. Indeed, we did find some colocalization of TM9SF4 with ezrin and actin stress fibers at the cell periphery (Figure S15). We have also added a full paragraph in the Discussion section to discuss this issue (page 15, Line 423-433).

3. The authors should comment all the data supporting the relationship between TM9SF4 and a pivotal activity of malignant cells such as cannibalism or cell-in-cell phenomena in general, as it has been commented in a recent perspective article (Fais S, Overholtzer M. Nat Rev Cancer. 2018). Of interest TM9SF4 has a homologous in amoebas and some analogies between cancer cells and amoebas have been proposed (Fais S, Fauvarque MO. Trends Mol Med. 2012).

Answer: We are grateful for this suggestion. We have added more comments on cannibalistic activity (amoebas, analogy between cancer cells and amoebas) in discussion (page 14, line 388-408). We believe that your suggestion greatly improved the manuscript.

4. A previous paper has shown that TM9SF4 is involved in the V-ATPase activity, in turn representing a key mechanism favouring the extracellular acidification of cancer cells (Lozupone et al Oncogene 2015). Actually, tumor cells live in acidic condition and many studies have shown that the same cell lines markedly change as cultured at different pH conditions (e.g. De Milito et al Int J Cancer 2010; Logozzi et al Cancers 2018; Urbanelli et al J Enzyme Inhib Med Chem. 2020). In trying to reproduce the in vivo condition the authors should perform new experiments comparing the various cell lines at least at the standard pH condition (7,4) and at the tumor pH (6,5). I guess it is worthy to be mentioned that tumors are acidic and that probably all the data obtained with tumor cells cultured

at pH 7.4 are not reliable, paradoxically a buffered pH may cure tumors (Pillai SR, et al. Cancer Metastasis Rev. 2019)

Answer: Thank you for your thoughtful comments. Indeed, acidic extracellular pH is a major feature of tumor tissue, primarily due to lactate secretion from anaerobic glycolysis. However, please note that in our experiments, A2780 cells are maintained in standard DMEM that was only weakly buffered with bicarbonate buffer without addition of HEPES. The medium pH was decreased rapidly during culture due to metabolic acid production, which could easily be visualized by media color change to orange color. A2780 grows very fast and has a high metabolic rate, therefore the culture medium quickly became acidic and the cells were actually growing at acid acidic condition most of their times.

The manuscript needs extensive English revision.

Answer: English is re-edited

Reviewer #2, Expertise: actin dynamics, cancer metastasis (Remarks to the Author):

In this manuscript by Meng et al. submitted to Nature Communications, the authors investigate the role of TM9SF4 in cell cancer cell behaviours and actin cytoskeleton dynamics. The authors provide evidences that knocking down TM9SF4 increases cell spreading, affect cell polarity and promotes actin filament assembly. These cellular defects are associated with a reduction in cancer cell abilities to migrate, invade and self-renew in vitro and to grow and to form metastasis in nude mice. Using a very elegant experiment, the authors also show that TM9SF4 is cleaved to produce a N-terminal fragment (NTF) with abilities to bind to F-actin to induce actin oxidation and to enhance cofilin-mediated F-actin disassembly. The authors propose that the cleaved NTF of TM9SF4 promotes cofilin-mediated F-actin disassembly to regulate cancer cell migration and metastasis. This work is interesting and the experimental data provided are globally well done. I have nevertheless a few concerns about some of the data, their interpretation data and the conclusion draw by the authors.

Majors points:

1. In some Western Blots, actin is used as loading control, as TM9SF4 controls actin dynamics, it could be wise to provide evidence that TM9SF4 does not affect total actin levels.

Answer: We appreciate the reviewer for this suggestion. Although TM9SF4 controls actin dynamics, the total amount of actin does not change. We have added a new figure to show that overexpression or knockdown of TM9SF4 did not affect total actin levels compared with the GAPDH control (Figure S1).

2. The authors claim that the number of actin stress fibres is increased in TM9SF4 knocked down cells (Fig. 1e). I could not find information on how stress fibres were evaluated but if they were evaluated based on Phalloidin staining only, this might not be sufficient to make such a claim. If the authors are referring to actin fibres crossing the cells, they could be ventral stress fibres but they could also be others actin structures, like TAN lines. Are the actin fibres identified by the authors attached to Focal Adhesions, are phospho-myosin II positive and found basal to the nucleus? Related to this point is the localization of Paxillin as a marker of Focal Adhesion in Fig. 5a. It would be important to verify that other Focal Adhesions markers show the same accumulation in dot-like structures (Zyxin? Vinculin?) because based on the image provided for cells knocked down for TM9SF4, it does not look like that the Paxillin punctuated staining are associated with actin stress fibres. It could be trafficking defect of Paxillin, which accumulates in vesicles.

Answer: We agree with the reviewer's point. Actin forms different stress fibers that vary in their location and in the attachment to focal adhesions. For example, ventral stress fibers and peripheral bundles lie at the basal side and their ends terminate in adhesions. Dorsal stress fibers are anchored at adhesions at one end only and they rise up toward the cell's dorsal side. Transverse actin arcs are not anchored in focal adhesions, and they are linked to dorsal fibers and run parallel with the leading edge. TAN lines are located above the nucleus and connected to the nucleus through the LINC complex.

In the present study, we counted the percentage of cells containing thick actin stress fiber with a diameter > 0.5µm. Therefore, all kinds of actin stress fibers were counted without differentiating different subtypes. For focal adhesion experiments, we took your suggestion and performed vinculin staining. The result showed that TM9SF4 suppression also resulted in increased vinculin expression (Fig. S14).

3. Conversely, the authors claim that overexpression of TM9SF41-258 caused a drastic disruption of F-actin fibres in NIH3T3 cells (Fig. 2h). From the picture provided, we can see that TM9SF41-258 induces a kind of clustering of F-actin, which could prevent the visualization of actin fibres. Does overexpressing the full length induce the same phenotype?

Answer: We thank the reviewer for this question. Full-length version does not have a similar effect. We explored the reason and found out that full length of TM9SF4 could not properly be cleaved to form N-terminal fragment in NIH3T3 cells. However, please note that in ovarian cancer cell A2780, full-length TM9SF4 can be cleaved to form N-terminal fragment (Figure S9), which can explain important function of TM9SF4 in ovarian cancer migration and metastasis.

4. TM9SF4 knocked down cells appear very round in Fig. 1c but form many extruding spikes in Fig. 1f. It makes it hard to conclude on the role of TM9SF4 on cell shape? Are cells in Fig. 1c representative of the phenotype?

Answer: We thank the reviewer for this insightful question. Control cells are regularly sized and round (no extruding spikes), whereas most TM9SF4 KD cells had a much large size with round shape. In addition, significant portion of TM9SF4 KD cells displayed abnormal actin spikes and long actin-stained filopodia, which agree with the previously published data about TM9SF4 mutant immune cells (Bergeret E, et al., J Cell Sci. 2008;121:3325-3334). Similar results were described in HEK293 cells following knockout of TM9SF4. TM9SF4 KO in HEK293 cells resulted in generation of both phenotypes including large round cells and cells with many extruding spikes (Perrin J, et al., J Cell Sci 2015;128:2269-2277).

5. Could the authors discuss how it could be that apolar cells (control cells) migrate faster than polar cells (cells knocked down for TM9SF9) that appear to form more striking lamellipodia as shown in Fig. 1c or 4g?

Answer: Reports show that some metastatic tumor cells, such as metastatic amoeboid MTLn3 tumor cells, display characteristics of apolar high turning frequency type of migration (Sidani M, et al., J Cell Biol 2007;179:777-791). In our study, metastatic ovarian cancer cells A2780 cells also displayed similar apolar high turning frequency type of motility. Knockdown of TM9SF4 reduced turning frequency of the cells, which agreed with its reduced metastatic ability. This is added to the Discussion part (Line 409-422).

6. In Extended data Fig. 1, the authors show that knocking down TM9SF4 reduces cell growth (the sum of proliferation and apoptotic events). In the text they claim that TM9SF4 promotes cell survival, while in the legend, they claim, it promotes proliferation. Are TM9SF4 knocked down cells die more or proliferate less? They also show that TM9SF4 knocked down cells are binucleated (Extended data Fig. 2). Could the decrease in migration, invasion and metastasis in nude mice due to the lack of cell

survival or binucleated phenotypes? This is important to conclude on the direct or indirect effect of TM9SF4 on cell migration.

Answer: We performed experiments to differentiate the various possibilities mentioned by the reviewer. Our data show that knockdown of TM9SF4 does not cause marked cell death (Figure S5). It only affects cell proliferation (Figure S4). Actin is involved in cell proliferation and division. TM9SF4 regulates actin dynamics. Therefore, it is not surprising that TM9SF4 can affect cell proliferation. Please note that in cell migration assay, we used standard protocol of minimal serum, in which the contribution from cell proliferation can be ignored. Therefore, the results only indicate direct effect of TM9SF4 on migration.

7. The GFP levels in mutation/depletion of the ABD is very weak and not concentrated in puncta (Extended data Fig.6), compared to the TM9SF41-258 in Fig. 2h. Are these mutation/deletion versions unstable? If not, the intensity of GFP should be the same for the 4 TM9SF4-GFP constructs. This is important to determine whether they co-localize with F-actin or not.

Answer: Difference in GFP level between (old Extended data Fig. S6) and new Extended data Fig. S10 is due to variation of individual cells rather than the effect of different molecular constructs. Because all these constructs were transiently expressed in cells, it's hard to ensure that each cell obtains equal amount of plasmids. Overall, we did not see any difference in GFP intensity with different constructs.

8. Based on the model provided, it would be important to provide a piece of data showing that cells expressing TM9SF41-258 (NIH3T3 for example) migrate/invade more in vitro or grow more/bigger spheres.

Answer: We thank the reviewer for this important question. We took your suggestions and performed additional experiments. The results showed that NIH3T3 cells could not grow into sphere when TM9SF4-NTF was overexpressed, probably due to lack of other pivotal proteins required for stemness. However, overexpression of TM9SF4_NTF indeed enhances the migration and invasion of NIH3T3 cells (Figure S16).

Minor points:

1. Although the text could benefit from some corrections, it is in general well written. However, one part that could be improved substantially to make the manuscript more appealing is the discussion. This part is mostly a repetition of the result part, with little discussions in the context of the literature. This is revealed by the fact that the first half of the discussion does not contain any reference. The model in Fig. 7 could be discussed in the context of the current knowledge.

Answer: The discussion part is substantially revised to reduce the repetition and to add the discussion related to literature and current issues.

2. The authors used different cell lines to perform this study but it is not always clear which cell line have been used for each experiment. For example, in Fig. 1f, are those A2780 or MCF7 cells? This is important to compare data with other panels using these cells. These information could be easily included in the legends

Answer: More information is added. Thank you

Reviewer #3, Expertise: actin dynamics, biochemistry (Remarks to the Author):

The manuscript "TM9SF4 is a novel F-actin disassembly factor that promotes cancer cell migration and metastasis", by Meng and colleagues, addresses the N-terminal part of TM9SF4 regulates actin dynamics. In particular, it highlights that this N-terminal fragment possesses and synergize with cofilin to tune actin turnover in cells. The paper is well written and a large array of experimental approaches allows to convincingly gain further insight into the activity of TM9SF4. Though I have a major concern about the molecular description that is made in the discussion section of the manuscript on how the NTF of TM9SF4 may cooperate with cofilin to induce actin disassembly in vitro: There is no proof that any actin conformational change caused by TM9SF4_NTF that would enhance cofilin disassembly activity. As explained in details below, I think the biochemical conclusion is not scientifically correct and the discussion should be modified to reflect the exact knowledge gained by the authors in this manuscript. Alternatively, additional experiments to derive more advanced molecular details should be designed and performed. In general, the method section for the in vitro characterization of TM9SF4 is not describing the experiments with enough details, especially buffer conditions and product concentrations.

Major modifications should therefore be made before the manuscript can be considered as scientifically valid.

Major points:

- figure 2a: signal peptide color should be modified to match between what is shown in the sketch and the diagrams. Also, please indicate the TMDx domains, as well as the amino-acid positions in the sequence.

Answer: Corrected. Thank you.

- line 234: I agree with the results shown figure 3j, that subtilisin cleavage is impaired in the presence of TM9SF4_NTF, but I disagree with the interpretation that TM9SF4_NTF induces an F-actin twist. First, in the method section, there is no indication whether the assay is performed in non-reducing condition or in the presence of GSH to induce glutathionylation. The binding of TM9SF4_NTF alone could protect actin from subtilisin digestion.

Answer: The experiments were performed in non-reducing condition. We agree with the reviewer's point that the impaired subtilisin cleavage in the presence of TM9SF4_NTF may not be sufficient to prove that TM9SF4_NTF could induce F-actin twist. Previous studies showed that enzyme subtilisin cleaves unoxidized actin between Met47 and Gly48 and does not cleave oxidized actin under limited proteolysis conditions (Grintsevich EE, et al., Nat Cell Biol 2016;18:876-885). Therefore, it can be used as an indicator to show whether actin is oxidized or not (Grintsevich EE, et al., Nature Cell Biol 2016;18:876-885). Thus, we now removed the wording "F-actin twist". Instead, we now only mentioned "F-actin oxidation".

- line 243, figure 4b: **The authors do not show that NEM treatment alone prevented the binding of cofilin, and therefore its depolymerizing activity.** This point has been addressed in the past by others (for example, by the Emil Reisler lab: <https://doi.org/10.1021/bi400715z>). As a consequence, the conclusion derived from this assay should be reassessed.

Answer: We thank the reviewer for raising these important points. We took your suggestion and conducted an experiment to see if NEM treatment alone could affect actin depolymerization. The

results showed that NEM itself had no/minor effect on cofilin-mediated-actin depolymerization (Figure S11).

- line 200: it is unclear if the affirmation made in the sentence refers to the article in the reference 26 or to the results shown in figure 3a.

Answer: We thank the reviewer for this comment. The affirmation made in the sentence refers to the article in the reference 26. We have rewritten the sentence to eliminate the ambiguity (page 8, line 219-220, new reference number 18).

- line 890: the method section lacks information about the buffer used for NADH consumption and NADH concentration.

Answer: Detailed information "a reaction mixture containing 20 mM Tris-HCl pH 8.0, and 200 μ M NADH at 37°C with stirring" was added (page 23, Line 655).

- **Are the authors able to exclude that oxidation is not happening on Met 44 and Met47 in the presence of NADPH?**

Answer: We thank the reviewer for raising these important points. To answer this question, purified TM9SF4_NTF were incubated with actin, followed by analysis of oxidation status at Met 44/47 by mass spectrometry. Supplemental Table 1 and 2 shows the mass spectrometer results, one with actin alone (Supplemental Table 1) and another with actin + TM9SF4_NTF (with NADPH) (Supplemental Table 2). The related peptide fragment is HQGVM⁴⁴VGM⁴⁷GQK (highlighted in red in Supplemental Table 1 and 2). 12 fragments of HQGVM⁴⁴VGM⁴⁷GQK can be found in both samples (highlighted in red in Supplemental Table 1 and 2). In both samples, among the 12 HQGVM⁴⁴VGM⁴⁷GQK fragments, 8 are oxidized (67%). There is no difference between these two samples. These results support that TM9SF4_NTF does not induce oxidation on Met44 and Met47.

- figure 7: the cofilin has a weird shape.... please redraw it with its known shape and size, relative to actin subunits.

Answer: Corrected. Thank you.

- figure 7: the data do not show that cofilin is specifically recruited onto oxidized actin subunits, as shown in the model shown in figure 7. Please avoid confusing description of the conclusion of the performed experiments in the model figure.

Answer: Figure 7 is redrawn accordingly. Thank you.

- line 384: the sentence 'Lastly and more importantly, **TM9SF4-stimulated F-actin disassembly was abolished by prevention of F-actin oxidation with DTT/NEM.** ' is not backed up by any data. This is a very important point. Unless I misread the entire manuscript, this is never shown. For me, one possible explanation for TM9SF4_NTF being able to increase cofilin-induced F-actin disassembly is pure binding competition for actin residues (eg. on Sub-domain 2 and/or sub-domain 1 where Cys374 is located). This binding competition is enough to explain data of figure 4 a, b and C. The cofilin-concentration dependence of F-actin disassembly is not straightforward (eg severing activity versus filament saturation by binding which prevents cofilin severing) and extra care should be made when trying to conclude from these types of assay. Furthermore, there is no indication in the method section that those assays were performed in non-reducing conditions or in the presence of NADPH or GSH, so the TM9SF4_NTF induced oxidation/glutathionylation might be minimal or even non-existent. This point should be discussed correctly or new experiments with the appropriate conditions should be performed to be able to derive the correct conditions. For example, the oxidation could be performed on F-actin by TM9SF4_NTF, the actin filament separated from TM9SF4_NTF by centrifugation before cofilin is added to induce depolymerization.

Answer: We thank the reviewer for this insightful question.

In Figure 4c, comparison of “actin + GST-TM9SF4_N24-258 + cofilin” with “actin + GST-TM9SF4_N24-258 + cofilin + NEM” clearly showed that NEM prevented the F-actin disassembly by cofilin and TM9SF4. Similarly, in Fig. 4b, four right columns showed that NEM prevented the conversion of F-actin (P) to G-actin (S) induced by cofilin and TM9SF4. We guess that you were asking whether we have evidence whether this is related to F-actin oxidation/glutathionylation. In this regard, Figure 3i showed that NEM and DTT could abolish TM9SF4-induced actin glutathionylation. Note that glutathionylation of F-actin occurs via oxidation of a cysteinyl residue to a sulfenic acid intermediate that readily reacts with GSH to form a mixed disulfide. Therefore, F-actin glutathionylation level can be used as an indicator of F-actin oxidation level.

To further confirm that F-actin disassembly by TM9SF4 (and cofilin) was related to F-actin oxidation/glutathionylation, we performed a new series of experiments. Site-directed mutagenesis was performed to replace Cys374 of actin with Ala. Cys374 is reported to be the highly reactive cysteine in actin and that it is the only site of actin glutathionylation (Dalle-Donne I, et al., Free Radic Biol Med. 2003;34(1):23-32; Lassing, I. et al. J Mol Biol. 2007;370:331-348). Cys374 is also known to be crucial for the maintenance of F-actin stability. Our results showed that the mutation at Cys374 abolished the TM9SF4_NTF-induced actin glutathionylation and nearly abolished the F-actin disassembly by cofilin and TM9SF4 (Figure S12-13). However, the mutation at Cys374 did not affect the binding between actin and TM9SF4 NTF (Figure S12). These data provided unequivocal evidence that TM9SF4-stimulated F-actin disassembly is related to F-actin oxidation/glutathionylation.

As for your speculation of binding competition between TM9SF4 and cofilin to actin, we believe that it is unlikely. Binding competition between TM9SF4 and cofilin should reduce the cofilin binding to actin, thus reducing the actin severing. However, in our experiments, TM9SF4 promotes cofilin-mediated actin depolymerization. In addition, the experiments in Fig. 4a-c were performed in non-reducing condition, which allows F-actin oxidation to occur. As for your last suggestion: “the oxidation could be performed on F-actin by TM9SF4_NTF, the actin filament separated from TM9SF4_NTF by centrifugation before cofilin is added to induce depolymerization”, in Fig. 4c and 4a we did have experiments comparing the effect of TM9SF4_NTF before and after cofilin addition. Nevertheless, as mentioned above, we have now performed a new Cys374 mutagenesis study, which provided the most compelling evidence for the involvement of F-actin oxidation in TM9SF4-stimulated F-actin disassembly. Herein, mutation at Cys374 abolished the TM9SF4_NTF-induced actin glutathionylation and nearly abolished the F-actin disassembly by cofilin and TM9SF4 (Figure S12-13).

minor points:

- ref 34 and 37 are the same.

Corrected

- typo line 166 GTP for GFP.

Corrected

-line 179: the statements on GFP and GST construct could be removed.

Corrected

- figure 3 legend, especially for the subsets c, d and e are not correctly mentioned in the legend and more details should be given.

Corrected

- line 862, I am guessing that the TM9SF4_N178-258 construct also contains a GST tag, like all the other constructs, right?

Yes, Corrected

-some actin reference literatures are sometimes a bit outdated....

Corrected

- line 347: 'it is clear that...' could be replace with something less direct, like 'our data strongly indicates'.

Corrected

- line 374, the reference to NADH consumption for oxidation is misplaced, as it would indicate that it refers to the activity of TM9SF family proteins.

Corrected

REVIEWER COMMENTS

Reviewer #1 (Remarks to the Author): Expert in ion channels

I remain on my previous suggestion this paper is in no way suitable for publication in NC due to the low level of originality of the study content

Reviewer #2 (Remarks to the Author): Expert in actin cytoskeleton

In their revised manuscript, Meng et al. answered most of my concerns. This is an interesting story with data of high qualities. I would nevertheless suggest to the authors to modulate the interpretation of their data in the discussion. In the discussion, the authors states: "...this study identified TM9SF4 as a novel protein that can promote the cofilin-mediated F-actin disassembly, thereby regulates cancer cell growth, migration and metastasis." A formal demonstration of this statement would be to show that introducing a mutation in the conserved leucine and lysine residues (LK), which directly interact with actin, by CRISPR prevents cell migration, invasion and tumour growth. These experiments are beyond the scope of this manuscript and as it stands, I recommend publications. Yet, the authors cannot formally exclude the possibility that the TM9SF4 NTF regulates part of TM9SF4 activity, while the cleaved membrane-associated TM9SF4 also has a role. Using a knocked down strategy, they see the sum of these effects. Some examples of such king can be found in the literature: FOXM1C for instance. Data argue that it is cleaved, giving rise to N-terminal domain, sufficient to prevent F-actin accumulation at the cortex during cell division, while it's C-terminus domain becomes transcriptionally active. The authors could be fully right at proposing that the only activity of TM9SF4 is through its NTF by regulating F-actin and discussing this point is critical. To sustain this model, they could discuss whether TM9SF4 KO phenocopies the one of Cofilin KO. Yet, the statement "...this study identified TM9SF4 as a novel protein that can promote the cofilin-mediated F-actin disassembly, thereby regulates cancer cell growth, migration and metastasis." is overinterpreted to me.

In Materials and Methods, the authors should indicate how they quantified stress fibers.

In Fig. S4, the authors should indicate what is each color. We guess red is EdU and blue DAPI but this has to be indicated.

In Fig. 5f, the authors should indicate which graph is for migration and which is for invasion.

In Fig. 5a, the authors should indicate which cancer cell line was used.

Reviewer #3 (Remarks to the Author): Expert in actin dynamics

I am satisfied with all the answers and improvements provided by the authors of the manuscript. I congratulate them for the effort put to highlight the molecular action of TM9SF4 to modify actin dynamics that has a direct impact on some cancer metastasis.

Still, I would appreciate 2 modifications to appear the upper right sketch in figure 7:

1. There is no evidence that TM9SF4_N remained bound to oxidized actin subunits upon oxidation. In fact the opposite is shown in the figure 4a gel. As a consequence, the authors should indicate that TM9SF4_N binds and unbinds from actin subunits.

2. It is now well accepted that cofilin binds cooperatively to filament side, form domains that will later fragment at their boundaries. I would therefore advise the authors to resketch how cofilin binds to filaments before fragmenting/disassembling filaments, oxidized or not.

Reviewer #4 (Remarks to the Author): Expert in ovarian cancer and metastasis

The manuscript entitled "TM9SF4 is a novel F-actin disassembly factor that promotes tumor progression and metastasis" by Meng et al. describes an impressive set of biochemical and cell biological assays to systematically investigate the mechanisms of TM9SF4 action in actin dynamics and cancer cell functions. To prove the biological importance of the described mechanism, the authors further test the effects of TM9SF4 knockdown in A2780 cells using functional assays from

cell migration and colony growth to tumor growth and metastasis in mice. The manuscript has been written in a clear way to help the reader through the extensive array of experiments and results. The amount of work is extensive and the results convincingly bring new mechanistic insight supporting an interesting function for cleaved TM9SF4 in actin regulation. My major comments are related to the functional validation experiments using the A2780 cells and the consideration of these results in association to ovarian cancer.

1. The A2780 cell line has been established from an ovarian endometrioid adenocarcinoma tumour. This is different from the major type of ovarian cancers, the high-grade serous ovarian cancer (HGSOC). HGSOCs represent not only the most common, but also the clinically most aggressive ovarian cancer type. While results from one cell line in general may not provide strong evidence for the significance of a given mechanism in the representative human disease, the results from A2780 cells certainly should not be generalized to ovarian cancer. In addition, lung metastasis are rare in ovarian cancer. Therefore, the final statement in Abstract, the second last sentence in Introduction (p. 4) as well as the consideration of the *in vivo* results in Discussion (p. 17) are unjustified as presented. These conclusions should be limited to the used cell model, not to ovarian cancer.

2. The functional validation experiments have been done in various ways, including migration, colony formation, and spheroid growth in cultured cells as well as in three different *in vivo* approaches using the A2780 cell line with and without TM9SF4 knockdown. However, additional results from some experiments using other relevant cell lines as well as rescue experiments with the central TM9SF4 construct, including the one coding for the N-terminal fragment, would help in proving that the cellular effects of the knockdown are specific to the mechanism identified in the manuscript.

3. Fig. 5: The presented results of the colony formation and cell spheroid assays could be explained largely by the reduced growth. Therefore, the result description could be limited more directly to what is measured. Alternatively, the stemness state of the cells could be characterized (and/or discussed). For the cell spheroid assay, the method description should include the number of cells used.

4. Could not find method description for the time-lapse microscopy for tracing of motility of the control cells and TM9SF4 knockdown cells (Fig. 5c-d) or that of the cytokinesis in Fig. S3. The description of these results is partly confusing. In the quantitative data, total path length did not differ but net path length and directionality were increased by the knockdown, whereas turning frequency was decreased. From these results the authors conclude that the knockdown decreased cell motility. Shouldn't the conclusion from this data be restricted to the finding that the cells changed from linear to random movement?

5. Fig S5 seem to show only results from one flow cytometry sample per treatment group, and thus lack summary data from replicates.

6. The results from Fig 2h-g are misleading. Fig. 2g should clearly mark that the samples compared in the western blot are from different cell lines and preferentially include detections of all the proteins used in 2h, including TM9SF4-GFP.

Point-to-point responses to Reviewers' comments

REVIEWER COMMENTS

Reviewer #1 (Remarks to the Author): Expert in ion channels

I remain on my previous suggestion this paper is in no way suitable for publication in NC due to the low level of originality of the study content

Answer: We thank the reviewer for the comment. Lozupone et al., reported that TM9SF4 affects cancer cell motility and migration (Lozupone et al., *EMBO Reports* 2009;10:1348-1354; Lozupone et al., *Oncogene* 2015;34:5163-5174). However, the previous studies did not investigate actin involvement. Our present study provides ample amounts of new findings, uncovering a novel mechanism of how TM9SF4 can regulate cofilin-mediated actin disassembly, thereafter, controls cancer cell motility and migration.

Reviewer #2 (Remarks to the Author): Expert in actin cytoskeleton

Comments: In their revised manuscript, Meng et al. answered most of my concerns. This is an interesting story with data of high qualities. I would nevertheless suggest to the authors to modulate the interpretation of their data in the discussion. In the discussion, the authors states: "...this study identified TM9SF4 as a novel protein that can promote the cofilin-mediated F-actin disassembly, thereby regulates cancer cell growth, migration and metastasis." A formal demonstration of this statement would be to show that introducing a mutation in the conserved leucine and lysine residues (LK), which directly interact with actin, by CRISPR prevents cell migration, invasion and tumour growth. These experiments are beyond the scope of this manuscript and as it stands, I recommend publications. Yet, the authors cannot formally exclude the possibility that the TM9SF4 NTF regulates part of TM9SF4 activity, while the cleaved membrane-associated TM9SF4 also has a role. Using a knocked down strategy, they see the sum of these effects. Some examples of such kind can be found in the literature: FOXM1C for instance. Data argue that it is cleaved, giving rise to N-terminal domain, sufficient to prevent F-actin accumulation at the cortex during cell division, while it's C-terminus domain becomes transcriptionally active. The authors could be fully right at proposing that the only activity of TM9SF4 is through its NTF by regulating F-actin and discussing this point is critical. To sustain this model, they could discuss whether TM9SF4 KO phenocopies the one of Cofilin KO. Yet, the statement "...this study identified TM9SF4 as a novel protein that can promote the cofilin-mediated F-actin disassembly, thereby regulates cancer cell growth, migration and metastasis." is overinterpreted to me.

Answer: Thank you for your thoughtful comments. We have tuned down our argument. Instead, we now stated: "This study identified TM9SF4 as a novel protein that can promote the cofilin-mediated F-actin disassembly. TM9SF4 may interact with cofilin to regulate cancer cell migration and metastasis" (page 14, 1st paragraph).

More importantly, we have now taken your suggestion and added more discussion on the issue of "TM9SF4 KO phenocopying the loss of cofilin in many aspects, including the increased formation of stress fibers and focal adhesions, and altered cell motility behavior in metastatic cells (page 17, 2nd paragraph). Some references about loss of colifin on cell motility behavior are also added (sidani et al., *J Cell Biol* 2007;179:777-791; Tahtamouni et al., *BMC Cell Biology* 2013;14,45)". In addition, we have also added the point that "we cannot

exclude the possibility that “the cleaved membrane-associated TM9SF4 also has a role” (page 17, 1st paragraph).

Comments: In Materials and Methods, the authors should indicate how they quantified stress fibers.

Answer: We analyzed the percentage of cells containing thick actin stress fiber with a diameter > 0.5 μm as an index for coarse quantification of stress fibers. The information is added in the Materials and Methods. Thank you.

In Fig. S4, the authors should indicate what is each color. We guess red is EdU and blue DAPI but this has to be indicated.

Answer: Corrected. Thank you.

In Fig. 5f, the authors should indicate which graph is for migration and which is for invasion.

Answer: Corrected. Thank you.

In Fig. 5a, the authors should indicate which cancer cell line was used.

Answer: We indicated that A2780 cells were used in the updated legend for Fig.5a. Thank you.

Reviewer #3 (Remarks to the Author): Expert in actin dynamics

I am satisfied with all the answers and improvements provided by the authors of the manuscript. I congratulate them for the effort put to highlight the molecular action of TM9SF4 to modify actin dynamics that has a direct impact on some cancer metastasis.

Still, I would appreciate 2 modifications to appear the upper right sketch in figure 7:

1. There is no evidence that TM9SF4_N remained bound to oxidized actin subunits upon oxidation. In fact the opposite is shown in the figure 4a gel. As a consequence, the authors should indicate that TM9SF4_N binds and unbinds from actin subunits.

Answer: Thank you for your comments. We have added that TM9SF4 can bind and unbind from actin in the upper panel of b in Fig 7 (also shown below, circled in red).

2. It is now well accepted that cofilin binds cooperatively to filament side, form domains that will later fragment at their boundaries. I would therefore advise the authors to resketch how cofilin binds to filaments before fragmenting/disassembling filaments, oxidized or not.

Answer: We thank the reviewer for raising these important points. Therefore, Figure 7 was re-sketched. The typical cofilin pathway with cofilin binding and disassembly at pointed end is illustrated in Fig 7. A (also shown below).

Reviewer #4 (Remarks to the Author): Expert in ovarian cancer and metastasis

The manuscript entitled “TM9SF4 is a novel F-actin disassembly factor that promotes tumor progression and metastasis” by Meng et al. describes an impressive set of biochemical and cell biological assays to systematically investigate the mechanisms of TM9SF4 action in actin dynamics and cancer cell functions. To prove the biological importance of the described mechanism, the authors further test the effects of TM9SF4 knockdown in A2780 cells using functional assays from cell migration and colony growth to tumor growth and metastasis in mice. The manuscript has been written in a clear way to help the reader through the extensive array of experiments and results. The amount of work is extensive and the results convincingly bring new mechanistic insight supporting an interesting function for cleaved TM9SF4 in actin regulation. My major comments are related to the functional validation experiments using the A2780 cells and the consideration of these results in association to ovarian cancer.

1. The A2780 cell line has been established from an ovarian endometroid adenocarcinoma tumour. This is different from the major type of ovarian cancers, the high-grade serous ovarian cancer (HGSOC). HGSOCs represent not only the most common, but also the clinically most aggressive ovarian cancer type. While results from one cell line in general may

not provide strong evidence for the significance of a given mechanism in the representative human disease, the results from A2780 cells certainly should not be generalized to ovarian cancer. In addition, lung metastasis is rare in ovarian cancer. Therefore, the final statement in Abstract, the second last sentence in Introduction (p. 4) as well as the consideration of the in vivo results in Discussion (p. 17) are unjustified as presented. These conclusions should be limited to the used cell model, not to ovarian cancer.

Answer: We agree with the reviewer that A2780 cannot represent HGSOC. To demonstrate whether our findings can be applied to HGSOC, we conducted a series of experiments on three commonly used HGSOC cell lines, namely, CaOV3, COV362 and OVCAR3. We also used another very commonly used ovarian adenocarcinoma line SKOV3, which is not classified as HGSOC (Fogh J and Trempe G. *New Human Tumor Cell Lines*. In: Fogh J., editor. *Human Tumor Cells In Vitro*. Springer; Boston, MA, USA: 1975. pp. 115–159). The results from these cells (new results) are very similar to those in A2780 (old results). Phalloidin staining assay revealed that knockdown of TM9SF4 in CaOV3, COV362, OVCAR3, and SKOV3 increased cell size and enhanced actin stress fiber formation in these cell lines (Fig. S2). We also showed that TM9SF4 knockdown in these cell lines reduced the cell population growth (Fig. S4), metastasis (Fig. S19), and colony/spheroid formation of ovarian cancer cells (Fig.S21). In addition, animal experiments suggested that TM9SF4 tumor knockdown nearly abolished tumor initiation and metastasis induced by CaOV3 in athymic nude (Fig.S22). Taken together, these results demonstrate that TM9SF4 also plays an important role in regulating actin dynamics and control cancer cell motility in HGSOC.

We agree that lung metastasis is rare for ovarian cancer. However, peritoneal metastasis is very common in ovarian cancer. Our data clearly showed that TM9SF4 knockdown completely abolished the peritoneal metastasis of injected ovarian cancer cells in mouse model (Fig. 6C-D, Fig, S22B). Nevertheless, as suggested by the reviewer, we still tuned down our argument and emphasized that the experiments were done in animal model.

2. The functional validation experiments have been done in various ways, including migration, colony formation, and spheroid growth in cultured cells as well as in three different in vivo approaches using the A2780 cell line with and without TM9SF4 knockdown. However, additional results from some experiments using other relevant cell lines as well as rescue experiments with the central TM9SF4 construct, including the one coding for the N-terminal fragment, would help in proving that the cellular effects of the knockdown are specific to the mechanism identified in the manuscript.

Answer: We thank the reviewer for the suggestion. We established stable TM9SF4 knockdown cell lines with commonly used HGSOC cell lines (CaOV3, COV362 and OVCAR3), and another commonly used ovarian adenocarcinoma line SKOV3. Functional validation experiments showed that TM9SF4 knockdown in all these cell lines reduced cell proliferation (Fig. S4), migration and invasion (Fig. S19), and colony/spheroid formation (Fig. S21). Importantly, animal experiments demonstrated that CaOV3-induced tumor initiation and metastasis in athymic nude were nearly abolished when TM9SF4 was knockdown (Fig. S22).

We are sorry that we did not do the rescue experiments, because such experiments will require stable expression of two different constructs into ovarian cancer lines, which is technically very challenging to us. We normally delivered the first molecular construct into a cell line by lenti-viral mediated transduction (stable transfection), followed by

LipofecAmine-mediated transient transfection of the second molecular construct. However, transfection efficiency for LipofecAmine-mediated transfection in ovarian cancer line is too low and cannot be sustained. Therefore, we were able to perform functional study with this strategy. Sorry about that.

Nevertheless, we believe that we have presented enough evidence for the importance of working hypothesis in many ovarian cell lines, especially in HGSOCs. Thank you for your understanding!

3. Fig. 5: The presented results of the colony formation and cell spheroid assays could be explained largely by the reduced growth. Therefore, the result description could be limited more directly to what is measured. Alternatively, the stemness state of the cells could be characterized (and/or discussed). For the cell spheroid assay, the method description should include the number of cells used.

Answer: In colony formation and cell spheroid assays, we analyzed the number of colonies and spheroid formed, but not the sizes of colonies and spheroids. We believe that, while the sizes of colonies and spheroids are more related to cancer cell growth, the number of colonies and spheroids may be more related to tumor initiation and stemness. Nevertheless, we agree that stemness needs to be characterized by multiple indexes, which we did not do. Therefore, we have taken your suggestion and removed the word “stemness’ in most manuscript texts, and only described “what was measured (spheroid formation and colony formation)”.

For method, we have now added that 1000 cells were used in the cell spheroid assay in the Materials and Methods section in the revised manuscript.

4. Could not find method description for the time-lapse microscopy for tracing of motility of the control cells and TM9SF4 knockdown cells (Fig. 5c-d) or that of the cytokinesis in Fig. S3. The description of these results is partly confusing. In the quantitative data, total path length did not differ but net path length and directionality were increased by the knockdown, whereas turning frequency was decreased. From these results the authors conclude that the knockdown decreased cell motility. Shouldn't the conclusion from this data be restricted to the finding that the cells changed from linear to random movement?

Answer: We thank the reviewer for raising this point. We have revised the description to make it clearer. Metastatic cancer cells often display “high turning frequency” or “random walking” motility, which allows them to change migration direction freely (Sidani, M. et al. 2007. The Journal of Cell Biology 179, 777-791), whereas directional motility is associated with non-metastatic cancer cells (Sidani, M. et al. 2007. The Journal of Cell Biology 179, 777-791). Similarly, in our study, metastatic ovarian cancer cells A2780 cells usually displayed a random walking motility. However, after TM9SF4 depletion, cell movement patten changed from “random walking behavior to “directional motility behavior”.

We have removed the wording of “decreased cell motility”, and now stated: “knockdown of TM9SF4 was found to change the cell movement patten from “random walking behavior” to “directional motility behavior”.

We have also added the method description of time-lapse microscopy, which were used to track cell motility as well as cytokinesis.

5. Fig S5 seem to show only results from one flow cytometry sample per treatment group, and thus lack summary data from replicates.

Answer: Summary data are now added. Thank you.

6. The results from Fig 2h-g are misleading. Fig. 2g should clearly mark that the samples compared in the western blot are from different cell lines and preferentially include detections of all the proteins used in 2h, including TM9SF4-GFP.

Answer: We are grateful for this suggestion. We have added the name of cell lines in the updated Fig. 2g and the corresponding legends. We have also clarified the treatment and detection of 2h in the corresponding legends.

REVIEWERS' COMMENTS

Reviewer #5 (Remarks to the Author):

The manuscript by Zhaoyue Meng et al. titled 'TM9SF4 is a novel F-actin disassembly factor that promotes tumor progression and metastasis' is a very well written, scientifically sound and appropriate conclusions made manuscript. The changes made to manuscript and experiments to address the authors responses are appropriate. However, I feel the scientific rigor and impact of these studies do not fit Nature Communications current standards. I would consider another journal for publication of this important study.

Point-to-point response to the reviewer #5:

Remark from Reviewer #5: The manuscript by Zhaoyue Meng et al. titled 'TM9SF4 is a novel F-actin disassembly factor that promotes tumor progression and metastasis is a very well written, scientifically sound and appropriate conclusions made manuscript. The changes made to manuscript and experiments to address the authors responses are appropriate. However, I feel the scientific rigor and impact of these studies do not fit Nature Communications current standards. I would consider another journal for publication of this important study.

Answer: We appreciate the reviewer's conclusion that "the manuscript is well written, scientifically sound and appropriate conclusions were made". More importantly, the reviewer #5 agreed that "we appropriately addressed the questions raised by reviewer #4".

As for the scientific impact, in this study we identified TM9SF4 as a protein that can promote the cofilin-mediated F-actin disassembly. Previously, two types of redox enzymes, NADPH oxidases (NOX) and Mical flavoprotein monooxygenases, are documented to regulate actin dynamics. Through this study, we found another family of redox proteins, TM9SF family, that can regulate actin dynamics. Importantly, we found critical physiological function of TM9SF4 in regulating ovarian cancer migration and metastasis. We believe that these findings are important enough to merit publication in "Nature Communications". Again, we thank the reviewer #5 for your comments.